# Intimate Relations—Mitochondria and Ageing

**DOI:** 10.3390/ijms21207580

**Published:** 2020-10-14

**Authors:** Michael Webb, Dionisia P. Sideris

**Affiliations:** Mitobridge Inc., an Astellas Company, 1030 Massachusetts Ave, Cambridge, MA 02138, USA; mikewebb64@gmail.com

**Keywords:** mitochondria, ageing, energetics, ROS, gene regulation

## Abstract

Mitochondrial dysfunction is associated with ageing, but the detailed causal relationship between the two is still unclear. We review the major phenomenological manifestations of mitochondrial age-related dysfunction including biochemical, regulatory and energetic features. We conclude that the complexity of these processes and their inter-relationships are still not fully understood and at this point it seems unlikely that a single linear cause and effect relationship between any specific aspect of mitochondrial biology and ageing can be established in either direction.

## 1. Introduction

The last two decades have witnessed a dramatic transformation in our view of mitochondria, their basic biology and functions. While still regarded as functioning primarily as the eukaryotic cell’s generator of energy in the form of adenosine triphosphate (ATP) and nicotinamide adenine dinucleotide (reduced form; NADH), mitochondria are now recognized as having a plethora of functions, including control of apoptosis, regulation of calcium, forming a signaling hub and the synthesis of various bioactive molecules. Their biochemical functions beyond ATP supply include biosynthesis of lipids and amino acids, formation of iron sulphur complexes and some stages of haem biosynthesis and the urea cycle. They exist as a dynamic network of organelles that under normal circumstances undergo a constant series of fission and fusion events in which structural, functional and encoding (mtDNA) elements are subject to redistribution throughout the network. These processes are intimately linked to the health of the mitochondrion and it would not be an exaggeration to state that the health of the cell is a reflection of the health of its mitochondria.

The endosymbiotic hypothesis for the origin of mitochondria from aerobic bacteria engulfed by the primitive eukaryotic cell was originally proposed by Sagan [1] and has now gained general support. During their 1.45 billion years of co-evolution, most of the endosymbiont’s genes have been transferred to the nucleus of the host cell and of the approximately 1500 proteins that constitute the mitochondrial proteome, only a minority are encoded by mtDNA. In mammals mtDNA encodes 13 proteins of the electron transport chain, 22 mitochondrial specific tRNAs and two ribosomal RNAs. This distribution of genes across two genomes clearly implies a co-ordinate regulation of the activities of nuclear and mitochondrial genes [2].

Mutations in mitochondrial DNA are causative of several disorders with specific clinical manifestations and declining mitochondrial function, reflected in defects in ATP synthesis and increased generation of toxic reactive oxygen species is a universal feature of natural ageing. It accompanies the other hallmarks of ageing which include progressive loss of function in multiple organs, sarcopenia and increasing maladaptive low-grade inflammation. These end in death, which is a cumulative result of loss of function, leading to either increased vulnerability to environmental hazards such as predation and disease or to failure of critical organ systems such as the heart, liver or kidney. Several processes that may contribute mechanistically to age related degeneration have been identified, including oxidative damage, accumulation of toxic protein aggregates, autoinflammatory processes, loss of stem cell populations and an increasing load of malfunctional senescent cells. Are these processes linked to declining mitochondrial function and if so, is there a cause and effect relationship? if so, in which direction does it operate?

Mitochondria generate reactive oxygen species (ROS) as a consequence of the electron transport that drives oxidative ATP synthesis. These have a dual aspect, functioning at low levels as an important communication system regulating multiple cellular responses but also directly damaging macromolecules to a degree proportional to their proximity to high ROS concentrations. Thus, mitochondrial components, proteins, lipids and carbohydrates, are all subject to accumulating ROS damage which impairs mitochondrial function. This impairment of function by damaged molecules is reduced by two mechanisms, (1) their redistribution and dilution by the same fission/fusion process that distributes newly synthesized components and (2) the selective elimination of mitochondria which have acquired sufficient damage by a macroautophagic process known as mitophagy.

Mitochondrial quality is therefore maintained by the linked process of biogenesis, dynamic fission/fusion and mitophagy and their importance is indicated by the fact that impairment of any of them is associated with poor cell function and viability. These processes operate in a time scale of minutes to hours and yet mitochondrial function nevertheless declines over decades with age. This can be seen in many different parameters including decreases in ATP, changes in the activity of electron transport and mitochondrial enzymes such as citrate synthase, reduced respiratory capacity, changes in substrate preference, increasing accumulation of mutations in mtDNA and loss of overall mitochondrial mass [3,4,5]. This raises the question of why mitochondrial quality control processes increasingly fail to compensate for age related declines in mitochondrial function and indeed some evidence suggests that these processes themselves become less efficient with age.

Simpler eukaryotes such as yeast and *C. elegans* have been invaluable in unraveling the molecular details of mitochondrial biology, such as dynamics and the mitochondrial unfolded protein response. However, as unicellular (yeast) and largely post-mitotic (*C. elegans*) organisms, they cannot represent the complexity of the mammal, with its differentiated cell types and organ systems, homeothermy and mix of mitotic and post mitotic cells. We will therefore refer to results from simpler organisms where germane but focus most of the discussion on data from mammals. We will consider here only the processes and mechanisms relating natural ageing to mitochondria and interventions to extend lifespan are not discussed. We have reviewed this area broadly and constraints of space have limited the detail in each section. However, more specialist reviews on each of the areas discussed have appeared in recent years and are readily found using appropriate search criteria. Figure 1 indicates some of the key themes to which we will refer.

## 2. Molecular Biology of mtDNA

The mammalian mitochondrial genome encodes a set of 22 mitochondrial specific tRNAs, 2 ribosomal RNAs and 13 polypeptides, all of which are components of the electron transport chain [6,7]. It is embodied in a circular DNA molecule of 16.5 kb diameter, which is present in copy numbers between 100 and 10,000 per cell, depending on the cell type [7]. If all of these are identical, the cell is genetically homoplasmic, the term heteroplasmy being used to describe a cell with variant mtDNA sequences. Such variation may arise from pre- existing mutations inherited through the germ line or they may arise spontaneously during the lifetime of the cell. Because of the background fission and fusion that constitutes normal mitochondrial dynamics, variant mtDNAs are likely to be shared throughout the network, rather than characterizing individual clones of mitochondria.

The two strands are designated heavy and light, based on their nucleotide composition and most of the coding sequences are on the heavy strand, the light strand of mammalian mtDNA encoding eight tRNAs and a single polypeptide. mtDNA displays an economic organization, having no introns and only a single noncoding regulatory region; both tRNA and rRNA are relatively smaller than their cytoplasmic counterparts. There are also several differences in codon use from the universal code, such that methionine is encoded by AUA (conventionally isoleucine), AAG and AGG are both stop codons (arginine in the universal code) and UGA encodes tryptophan (termination in standard code) [8]. mtDNA is organized as DNA protein complexes (nucleoids) which have a looser structural organization of the homologous chromatin of the nuclear DNA. Nucleoids are located preferentially at sites of mitochondrial/endoplasmic reticulum contact and their location predicts the sites of future mitochondrial fission [9]. They contain several binding and packaging proteins, of which the most prominent is the double box high-mobility group DNA-binding *Transcription Factor A, Mitochondrial* (TFAM). Newly synthesized RNA is concentrated in RNA granules, adjacent to nucleoids and enriched in the DNA helicase Twinkle and single strand DNA binding protein mtSSB [10].

The replication of mtDNA has several unique features that distinguish it from nucDNA replication. As mentioned above, there is one non-coding regulatory region (NCR) which harbors promoters for polycistronic transcription from both the light and the heavy strand and one origin of replication for the heavy strand. A separate origin of replication lies outside the NCR. DNA synthesis is carried out by a dedicated DNA polymerase with proof reading capability, Polγ. In human, this is a heterotrimeric enzyme composed of a catalytic subunit (A) and two accessory subunits (B). Several additional polymerases have been identified (PrimPol, DNA polymerase b, DNA polymerase θ and DNA polymerase ζ) which play non overlapping roles with Polγ. Their functions are not completely elucidated but they are non-essential for mtDNA maintenance and are possibly involved in DNA repair processes.

Mutation rates of mitochondrial DNA are estimated to be approximately 10–70-fold higher than nuclear DNA [11,12]. This high rate has been attributed to several different causes. mtDNA lies near the sites of reactive free radical generation, exists in more vulnerable single stranded form for much of the replication cycle and unlike nDNA, is not highly compacted into more protected chromatin. However, the emerging consensus is that in spite of this, the major source of mutation in mtDNA is copying errors introduced during replication [12,13]. Ziada et al. [13] measured rates of mtDNA point mutations in the D loop of human patients between the ages of 2 and 72 years. They measured both transitions (thought to arise from Polγ errors) [14,15] and transversions (thought to arise from oxidative damage) [16] and found that both increased with age but transition mutations were more prominent, suggesting a greater contribution of polymerase errors to accumulation of mtDNA mutations with age. The evidence for replication errors as the major source of mtDNA mutation has been well summarized by DeBalsi et al. [17]. Although it was originally thought that mitochondria lacked an efficient DNA repair system [18], it is now clear that repair can take place by several mechanisms including base excision and replacement, repair of single and double strand breaks and mismatch repair [19]. Although these repair mechanisms can eliminate newly arising mutations, it should be noted that some repair processes are error prone and can potentially introduce new mutations including single base changes and deletions.

## 3. Regulation of Mitochondrial Gene Expression and Age-Related Changes in Mitochondrially Encoded Proteins

Of the approximately 1500 proteins constituting mitochondria, only a small number (13 in human) are encoded in mitochondrial DNA and translated on mitochondrial ribosomes. Mitochondrial biogenesis requires a close co-ordination of the synthesis, import and localization of the nuclearly encoded cytoplasmically synthesized mitochondrial proteome with the mitochondrial endogenous protein synthesis system. Figure 2 indicates some of the ways in which bidirectional communication between nucleus and mitochondrion is achieved. In the following discussion, we will refer to mitochondrial translation and mitochondrial translation products to indicate the process and products of intra-mitochondrial synthesis on mitochondrial ribosome; “mitochondrial proteins” will refer to all protein components of mitochondria, whether nuclearly encoded and cytoplasmically synthesized or mitochondrially encoded and synthesized.

Significant progress in describing the overall mitochondrial proteome has been made in recent years [20,21] and this has identified not only proteins that physically interact but also groups of proteins that are co-regulated [22] paving the way to link mitochondrial proteomics with normal and pathophysiological regulation. Two key observations are that a large fraction of the mitochondrial proteome is expressed in a tissue specific manner in mouse [23] and that global mitochondrial proteomic changes are observed during ageing [24,25]. Down regulation of genes associated with electron transport activity is one common theme in ageing [26] and the result of reduced expression of newly synthesized mitochondrial components may be exacerbated by an increasing inability of the ageing cell to rid itself of old and damaged mitochondria [27]. Of the 13 mitochondrial translation products examined by Short et al. [24] in normally ageing human, nine were significantly lower in older subjects, the declines in COX3 and COX4 being roughly of the same order of magnitude as the decline in ATP synthetic capacity. This was assessed using substrates for complex I (glutamate, malate) and complex II (succinate in the presence of rotenone) and indicated that the synthesis per g of tissue declined by about 8% per decade or 5% if the data were expressed per mg mitochondrial protein. They also showed that in normally ageing human, mtDNA and mtRNA abundance declined together with mitochondrial ATP production. In a separate study, quantitative mass spectrometry in mitochondria isolated from the brain of 5, 12 and 24 month old mice revealed changes in the expression of proteins involved in catabolic pathways and generation of metabolites, further highlighting that age-specific metabolic demands may influence the patterns of protein synthesis [25]. The induction of senescence (discussed more fully later) has also been linked to deterioration of mitochondrial OXPHOS [28,29,30].

An accumulation of mtDNA mutations with age or age-related disorders has been found in a variety of human tissues, including skeletal muscle [31,32] and brain [33,34,35] and heart [36]. All of the 13 polypeptides encoded by the human mitochondrial genome are subunits of the respiratory chain complexes, which in every case also contain subunits encoded by nDNA. Complex I is the first of the ETC complexes and catalyzes the transfer of electrons from NADH to ubiquinone. It contains the highest proportion of mtDNA encoded polypeptides of any complex, seven in the case of mammals. Mitochondrial genes ND1-6 are subunits of complex I, cytochrome b is the sole mitochondrial component of complex III, three cytochrome oxidase subunits of complex IV are encoded by genes Cox1-3 and ATP6 and ATP8 are subunits of the ATP synthase complex V. Complex II (succinate dehydrogenase) alone contain no mtDNA encoded subunits. The 12S and 16S ribosomal RNAs and 22 mitochondrion specific tRNAs complete the components encoded by mtDNA. Point mutations and deletions can occur in all of these genes and more than 300 such defects have been described [37]. Known genetic variants haplotypes have also been associated with specific components of the ETC, for example, the N haplogroup, the most common in Australia, encodes an allele of the ND3 gene, a subunit of complex I (m.10398A, ND3:114Thr) which is associated with increased ROS production [38].

We will now consider a few examples of well characterized mitochondrial mutations and their effects on energetic metabolism. As noted above, both complex I and complex III contain mitochondrially encoded subunits. These two complexes are the main sites of generation of free radicals. Failure of either complex can result in the production of a superoxide radical (O_2_.^−^). Superoxide can be detoxified by conversion to hydrogen peroxide by either superoxide dismutase and superoxide reductase [39], hydrogen peroxide being further converted to water and oxygen by catalase or glutathione peroxidase. A frameshift mutation in ND5 (complex I) identified by its resistance to the electron transport inhibitor rotenone was subsequently found to be identical to a defect in colorectal cancer lines [40]. Cell lines bearing this mutation had lower rates of ATP synthesis and higher levels of mitochondrial superoxide, although cytoplasmic hydrogen peroxide was not elevated, a result of an adaptive increase in expression of antioxidant enzymes. The decline in mitochondrial function and sensitivity to additional oxidative stress were dependent on the degree of heteroplasmy in a series of mouse cybrids carrying different proportions of a nonsense mutation in ND5. Mutations in other subunits of complex I are associated with Leber’s hereditary optic neuropathy (LHON) and display increased ROS generation [41,42].

Only one subunit of complex III, cytochrome b, is encoded by mtDNA. Nonsense, missense or frameshift mutations within the cytochrome b gene (MTCYB) are associated with complex III deficiency in muscle and a clinical presentation involving exercise intolerance [43]. In a longitudinal study of the NMRI mouse model of normal ageing, Reutzel et al. [44] measured brain mitochondrial function, cognitive performance and molecular markers every 6 months until mice reached the age of 24 months using 3-month-old mice as normal controls. At 6 months, expression of several mitochondrially related genes (complex IV, creb-1, β-AMPK and Tfam) were significantly elevated but mitochondrial respiration was reduced already by 12 months and ATP levels by 18 months. Similar findings on mitochondrial respiration were reported earlier in ageing primate [45] and rat brain [46]. Reduced mitochondrial membrane potential was seen but only in 24-month mice. Other studies in several tissues of C57bl/6 mice showed only a modest change in the ratios of activities of different components of the respiratory chain [47], although this would still likely result in reduced efficiency of the ETC. A potentially important observation in this study was that activities of the complexes were more adversely affected in tissues such as brain, heart and skeletal muscle, whose parenchyma is composed of postmitotic cells, than those in the liver and kidney, which are composed of slowly dividing cells. Decreases in expression of markers associated with mitochondrial biogenesis such as PGC1α, Nrf1 and TFAM were found in the livers of ageing rats [48] which also displayed reduced mtDNA content.

Impaired ability to switch between glycolysis and efficient OXPHOS may also result from mutations in mtDNA and these may affect cell fate decisions and the induction of senescence. This is discussed in greater detail in Section 7 but here we note that stem cells are typically glycolytic with a low abundance of mitochondria [49] and several types of adult stem cell including neural and hematopoietic require a shift to OXPHOS in order to differentiate [50,51]. A declining ability of mitochondria to carry out efficient OXPHOS or quality control through dynamics and the mtUPR may therefore have an impact on one key aspect of ageing, the inability effectively to replenish stem cell pools and produce specialized post mitotic cell types in various organ systems [52].

The induction of senescence has also been linked to deterioration of mitochondrial OXPHOS [53]. The four respiratory complexes of the electron transfer chain are composed of multiple protein subunits but the use of non-denaturing gel technology indicated that these themselves may be organized in higher order structures known as supercomplexes [53,54,55,56]. These observations have been confirmed in several more recent studies in which the structures of such complexes have been solved at high resolution [57,58,59]. The physiological significance of supercomplex assembly has been debated [57] and has been reviewed recently by Baker et al. [58]. Changes in the composition of the supercomplexes with age were described by Gomez et al. [59] who showed that most supercomplexes exhibited some decrease with age in rat heart—the larger complexes presenting the greatest decrements—while Frenzel et al. made similar observations in ageing rat cerebral cortex [60]. However, other studies found either opposite changes (an age related increase in I/III/IV supercomplex in ageing rat skeletal muscle [61] and O’Toole et al. [62] found no age associated change in supercomplex formation in ageing rat kidney. The precise physiological significance of the supercomplex is still under discussion (original suggestions include enhanced efficiency of electron transfer, substrate channeling, structural roles in the inner mitochondrial membrane or provision of a reserve pool of inactive respiratory complexes ready for activation when needed) and has been reviewed recently [63]. They were also proposed to play a role in limiting ROS production [64] in an hypothesis that linked the free radical theory of ageing to an increase in ROS production arising from supercomplex destabilization. Furthermore, the role of lipids, notably cardiolipin, in maintaining supercomplex structure has been noted (e.g., by Frenzel et al. [60] and if (as suggested later) cardiolipin is particularly subject to alteration by peroxidation, this would link ROS production, lipid peroxidation and supercomplex disruption with consequent accelerated ROS generation into a vicious cycle of increased damage and decreased mitochondrial function with ageing [65].

Overall rates of mitochondrial translation have been measured in vivo in human skeletal muscle and reported to decrease by approximately half between the ages of 24 and 54, with little further change thereafter [66]. The opposite was found by Miller et al. [67] who reported an increase in rat skeletal muscle mitochondrial translation with age. However, the rate of mitochondrial translation is rapidly responsive to changing energy demands and nutrient status [68] and factors such as exercise, diet and overall physiological status. Rates of mitochondrial translation should therefore be considered together with rates of turnover and the status of the mitochondrial proteostasis system, to be discussed later. It is conceivable, for example, that a higher rate of synthesis may be correlated with higher rates of clearance of aged and damaged mitochondrial proteins, confounding the expectation of lower synthetic rates with age. Shekar et al. [69] found that under conditions of heart failure, which is associated with reduced mitochondrial function, the turnover rate of several proteins involved in fatty acid oxidation, electron transport chain and ATP synthesis was increased. On the other hand, even mild heat stress causes reduced rates of mitochondrial protein synthesis, which is correlated with aggregation of the essential mitochondrial translation factor Tufm [70]. This aggregation and inactivation was interpreted as a protective response—by reducing mitochondrial protein synthesis, the accumulation of other aggregated mitochondrially synthesized proteins under stress conditions is avoided and this makes the point that changes in rates of synthesis or abundance of some proteins as a result of stress or ageing are not necessarily indicative of degenerative change.

In yeast, the accuracy of mitochondrial translation was shown to be a critical determinant of lifespan [71]. Mutations reducing the fidelity of translation in the conserved accuracy center of the yeast mitoribosome shortened chronological lifespan, while mutations leading to enhanced fidelity extended it. The effects of these mutations on lifespan is due to a coordinated quality control network integrating the mitochondrial protein synthesis with a cytoplasmic translational stress response which also included mechanisms dealing with aggregated proteins. However, mammals behave quite differently. Ferreira et al. [72] generated error prone and hyper accurate yeast and mouse cells and found that for the mammal, the rate of translation is more important than accuracy. Increased accuracy of mitochondrial translation slowed the rate of synthesis without inducing a compensatory stress response. This had a severe impact on mice carrying the mutation, which developed cardiopathy. An increased rate of mitochondrial mistranslation induced mitochondrial stress signaling, which accelerated mitochondrial biogenesis; it also affected telomerase expression and cell proliferation and the net result was to normalize metabolism.

The effects of heteroplasmy (multiple mitochondrial genomes within the somatic cells of an organism) are discussed more fully below but relevant to the need to for efficient co-operation between mitochondrially encoded subunits of the ETC and nuclearly encoded genes is the study of Sharpley et al. [73] which demonstrated that when mice containing a mixture of mt DNA from two different strains (129 and NZB mtDNAs, which differ in 91 nucleotides including 15 missense mutations) was produced, the two mt genomes increasingly segregated from each other in subsequent generations to produce mice with pure (homoplasmic) mtDNA from either source or a heteroplasmic mixture from the two parental strains. The heteroplasmic mice exhibited reduced oxidative phosphorylation reserve capacity, accentuated stress responses, cognitive impairment and reduced activity compared with their homoplasmic counterparts. This indicates that heteroplasmy even of two normal mitochondrial genomes without deleterious mutations can produce adverse physiological effects and provides a reason that uniparental inheritance of mtDNA may have evolved. The molecular reason for the impaired fitness of heteroplasmic mice is less clear but the authors suggest that it may be related to the need for interactions between different components of the electron transport chain, which have co-evolved for maximum complementarity, deviations from which are likely to impair function.

A connection between life span and the nuclear-encoded gene for EXD2 (exonuclease 3′5′ domain-containing 2), which has roles in modulating the aberrant association of mitochondrial messenger RNA with the mitochondrial ribosome was shown by Silva et al. [74]. Loss of EXD2 resulted in developmental delays and premature female germline stem cell attrition, with reduced fecundity. Counterintuitively, this was associated with a dramatic *increase* in lifespan, which could be reversed by antioxidant treatment. We will return later to a discussion of the relationship between ROS and lifespan.

## 4. Mitochondrial Biogenesis

While rates of mitochondrial translation have been taken as measures of mitochondrial biogenesis, the vast majority of mitochondrial components are encoded in nuclear genes and organelle biogenesis requires coordination of the rates of synthesis and import of components from these two compartments [75]. This co-ordination involves both temporal and spatial regulation—the different supercomplexes of the ETC, with their mix of mtDNA and nDNA encoded components are assembled at spatially distinct locations in the inner membrane [76]. In addition to providing the protein components of the mt translation machinery, nDNA encoded gene products are required for the processing of multigenic mtDNA transcripts [77]. Co-ordination of protein synthesis in the two compartments is under the control of nuclearly encoded genes and is sensitive to bidirectional signaling [72]. miRNAs have been implicated in this regulation in both anterograde and retrograde directions; miR-1 increases mitochondrial translation during muscle differentiation [78], mitochondrial COX1 mRNA expression is increased by nuclear-encoded miR-181c [79] and miR-663 controls mitochondria-to-nucleus retrograde signaling [80]. The exonuclease Myg1 was recently identified as a key component in the co-ordinate regulation of the nuclear and mitochondrial translation machinery [81]. It is located in both the nucleolus (where it regulates cytoplasmic translation) and in the mitochondrion, where it processes 3′-termini of the mito-ribosomal and messenger RNAs, thereby regulating translation of mitochondrial proteins. Failure to co-ordinate nuclear and mitochondrial translation may result in the formation of malfunctional and pro-oxidative enzyme complexes [82,83]. It is one possible mechanism by which defective mitochondrial biogenesis may be coupled with normal or pathological physiological ageing [84,85].

A simple reduction in mitochondrial mass in tissues such as skeletal muscle may underlie at least some of the age-related deficiencies in cellular energetics. However, loss of mitochondrial mass is not easy to quantify. Earlier morphometric studies indicated a loss of mitochondria from aged human hepatic cells [86] and mouse liver and heart [87]. The total area or volume occupied by mitochondria in electron micrographs may not be the most relevant measure of declining mass; El’darov et al. [88] showed that the area of the inner mitochondrial membrane per unit volume of mitochondria declined with age in two different rat strains, which has obvious functional implications as the electron transport complexes are assembled on this membrane. Quantitation of mitochondrially expressed but nuclearly encoded proteins such as citrate synthase has been used as an indicator of mitochondrial mass [89].

Although a basal level of mitochondrial synthesis is found in all tissues, mitochondrial biogenesis is intimately linked to the physiological status of the organism and transcriptional control of the formation of new mitochondrial components is regulated by several interacting pathways which are indicated in Figure 3. These are sensitive to such physiological stimuli such as nutritional status, exercise, hormonal status and the rhythms of the biological clock [90,91,92]. At a molecular level, there is extensive cross talk between the elements of these regulatory pathways, with examples of autoregulatory, feed forward and feed backward loops. One key question is the extent to which age-related changes in the activities of the genes responsible for mitochondrial biogenesis and function are responsible for the decline in mitochondrial function with age.

The expression of genes contributing to mitochondrial biogenesis requires transcriptional activators and co-activators. The former can bind recognition sites in the promoters of target genes and initiate transcription. The major transcriptional activators of mitochondrial biogenesis programs so far identified are Nrf-1 and Nrf-2 (also known as GA binding protein, GABP), Estrogen-related-receptors (ERR—α, β and γ), CREB, FOXO, PPARδ and cMyc [93,94,95] and their regulation by physiological stimuli is indicated in Figure 3. Their targets may include both mtDNA and nDNA encoded genes. Thus, Nrf1 initiates transcription of components of the OXPHOS system, some of which are mitochondrially encoded [96] and nuclearly encoded genes for mitochondrial proteins such as transporters and elements of the mitochondrial transcription/translation system including Tfam and mitochondrial ribosomal proteins [96]. It is regulated positively by interactions with PGC1α and negatively by interaction with cyclin D1 [96]. The related transcription factor Nrf2 has overlapping but not identical patterns of gene induction [97]. Nrf1 knockout mice are embryonic lethal and fail to maintain mtDNA [98] while Nrf2 knockouts are viable but exhibit impaired antioxidant defenses [99]. This suggests that a basal level of Nrf1 is required for normal mitochondrial biogenesis but this is enhanced in response to physiological signals triggering accelerated biogenesis [100,101,102].

The group of estrogen related receptors comprise three related nuclear receptors, ERRα, ERRβ and ERRγ of which the best studied is ERRα, which regulates the transcription of proteins involved in electron transport, the TCA cycle and mitochondrial dynamics [103,104]. In contrast to Nrf1, ERRα knockout mice are viable and fertile but show reduced ability to deal with exercise induced stress [105], suggesting that either other members of the receptor family can compensate for the loss of ERRα or that its functions are unnecessary for basal mitochondrial biogenesis. ERRγ knockouts have a more severe phenotype, display cardiac defects and die perinatally [106].

The activities of these transcription factors are coordinated by transcriptional co-activators, of which the best studied is PGC1α, together with its related coregulators PGC-1β and PRC. Positive and negative regulators of mitochondrial protein synthesis that are responsive to such physiological factors as exercise, hormonal status and nutritional status converge on PGC1α, commonly referred to as the master regulator of mitochondrial biogenesis. It is subject to post-translational activation by a wide array of upstream elements including 5′ AMP-activated protein kinase (AMPK), glycogen synthase kinase-3 (GSK-3) and Sirtuin1 (SIRT1). Collectively, these transduce signals related to the physiological status and energy demand of the organism. Various physiological signals activate mitochondrial biogenesis, including exercise, nutritional status (including caloric restriction), cold, hormonal status and the circadian clock. Thus, AMPK acts a sensitive indicator of the ratio of cellular ATP: (AMP/ADP), becoming activated in conditions of ATP depletion, whether caused by nutritional insufficiency such as caloric restriction or depletion of ATP due to exercise. It phosphorylates and activates Sirt1, which in turn deacetylates and activates PGC1α, resulting in increased transcription of mitochondrial genes [107,108]. During endurance exercise, AMPK activation of PGC1α is supplemented by PKA/CREB transcriptional upregulation consequent upon PKA activation occurring as a result of sympathetic activation of β-adrenergic receptors. ERRβ signaling is also upregulated in endurance exercise via upregulation of its ligand neuregulin [109,110] and contributes to an increased expression of the mitochondrial proteome. The sympathetic nervous system responds to cold by an increase in α-adrenergic signaling, which again leads to PKA phosphorylation and PGC1α upregulation via P38 [111].

These examples should be sufficient to illustrate the generalization that transcriptional control of mitochondrial genes is highly responsive to changing conditions and involves an integration of signals from many pathways. In addition, it can be tissue specific and involve autoregulatory feed forward or backward loops-ERRα and PGC-1α induce their own expression and in addition to activating ERRα GABP and NRF-1, PGCα also increases their expression levels. ERRα enhances the expression of both GABP and the negative regulator RIP140.

We return to the key question of whether there is evidence for dysregulation of this complex network with ageing and the extent to which this could account for decreasing mitochondrial function with age. We have already discussed the down regulation of genes associated with electron transport as a one common theme in ageing (Section 2 above) and the result of reduced expression of newly synthesized mitochondrial components may be exacerbated by an increasing inability of the ageing cell to rid itself of old and damaged mitochondria [27]. The mitotic status of the tissue under study may also be relevant, as long lived post mitotic cell types which are not replaced are more at risk from accumulation of defective mitochondria if they cannot be eliminated. In a study of ageing skin fibroblasts, Kalfalah et al. [112] found down-regulation of mitochondrial genes and corresponding decreases in mitochondrial content as the most prominent changes with age. This was associated with a (compensatory?) upregulation of AMP), increased PGC1α mRNA levels and decreased levels of Sirt1. Although aged cells still responded to pharmacological AMPK stimulation with induction of mitochondrial gene expression, the PGC1α-independent mitochondrial biogenesis response to starvation was attenuated and accompanied by increased ROS-production. Reduced AMPK activity has been reported in aged animals [113] and is directly linked to age-related insulin resistance and impaired fatty-acid oxidation [114,115,116]. This makes the point that an overall reduction in mitochondrial function need not imply a specific primary defect in mitochondria but rather a failure in the integration and transmission of the upstream physiological monitoring systems.

Relevant to this question is the finding of a transcriptional signature associated with caloric restriction, which delays age-related degenerative changes [117]. This signature includes genes in the functional category of mitochondrial energy metabolism and is conserved across species in *Drosophila*, rodents and rhesus monkeys. The same transcriptional signature was also seen in mouse genetic models of retarded ageing. It appears to involve Sirt3 signaling as a critical element and sirtuins are emerging as a central locus of the age-related decline in expression of the mitochondrial transcriptomes [118,119,120]. The reduction in sirtuin activity with age is likely linked to the age-related reduction in NAD levels discussed above and thus plausibly links the two processes with changes in fundamental aspects of mitochondrial biology [121].

## 5. Biochemistry of Mitochondrial Ageing

### 5.1. Small Metabolites

Changes in levels of several key metabolites are seen with age. However, in what follows, it is important to bear in mind that some changes may be more evident in some species than others and may also be tissue specific.

Coenzyme Q (2,3 dimethoxy-5-methyl-6-decaprenyl-1,4-benzoquinone; CoQ10, ubiquinol), is a small lipophilic molecule that is found in all biological membranes. It plays a key role in the inner mitochondrial membrane, where it functions as an electron acceptor and donor, transferring electrons between complexes I/II and III and interconverting between three forms, from fully oxidized ubiquinone, through partially oxidized semiquinone to fully reduced ubiquinol [122]. CoQ is linked to mitochondria not only by its key roles in electron transport and anti-oxidant activity within the organelle but also because the terminal steps of its biosynthesis are carried out within mitochondria by an enzyme complex of at least 12 proteins [123]. Its ability to function as an electron acceptor allows CoQ to function as an antioxidant, protecting membranes from oxidative damage and also recycling other antioxidant molecules such as ascorbate and a-tocopherol [124]. Levels of Coenzyme Q decline with age in some but not all tissues in both rodents [125] and human [126].

In addition to forming a key component of the body’s anti-oxidant system, CoQ may reduce the activation of NfκB by free radicals, thereby acting as a brake on chronic inflammation which is progressively relaxed as CoQ levels decline with age [127]. Some genetic defects in CoQ synthesis lead paradoxically to an increased life span [128] which is possibly due to mitohormesis, where production of a certain level of ROS initiates upregulation of antioxidant defenses to a degree that the overall benefit is greater than the damage sustained by the increased ROS [129].

The Krebs cycle (citric acid cycle, tricarboxylic acid cycle) is a series of enzymes localized in the mitochondrial matrix which catalyze successive stages in the conversion of acetyl CoA (itself derived from nutrient carbohydrate, fats and proteins), into ATP and carbon dioxide. It is a central component of many other biochemical pathways and in addition to ATP, generates reducing equivalents in the form of NADH and FADH2 and some amino acid and lipid precursors. Its major steps are indicated in Figure 4.

The Krebs cycle has also been proposed as an important component of anti-oxidant defenses via the formation of 2-oxoglutarate (also known as α-ketoglurate) [130]. In addition to its anti-oxidant roles, 2-oxoglutarate is also a substrate for 2-oxoglutarate-dependent dioxygenases (2-OGDO) [131]. These constitute a family of enzymes regulating DNA and histone methylation, including Ten-Eleven Translocation (TETs) and Jumonji C domain containing (JmjC) demethylases and they are inhibited by two other Krebs cycle intermediates, succinate and fumarate. The regulation of enzymes controlling epigenetic changes in DNA provides a potential link between age related changes in mitochondrial Krebs cycle activity with changes in nuclear gene expression that may contribute to the ageing process [132].

Aconitase catalyzes the stereospecific dehydration-rehydration of citrate to isocitrate, the first step in the Krebs cycle. It is subject to regulation by redox status as its catalytic activity is regulated by reversible oxidation of the iron-sulphur cluster [4Fe-4S]^2+^. Age dependent reductions in aconitase activity in rat heart and liver were described by Delaval et al. [133] and in kidney mitochondria of mice by Yarian et al. [134]. The latter compared young, middle aged and old mouse kidney and found that of all the Krebs cycle enzymes, aconitase showed the most significant decrease of activity with age. Although not considered to be the rate limiting step of the Kreb’s cycle, such functional decreases in aconitase activity may contribute to the overall decline in energetic efficiency with age.

Krebs cycle intermediates may play additional roles in signaling mechanisms relevant to some disorders of ageing. Thus, succinate is a ligand for a G protein coupled receptor, GPR91 and a related receptor GPR99 recognizes oxoglutarate [135]. Succinate increases blood pressure, an effect not seen in GPR91 knockout mice and there is therefore a plausible link between alterations in Krebs cycle intermediates and some disorders associated with ageing such as renal hypertension and atherosclerosis [136]. GPR91 is also expressed in ganglion cells of the retina. During the development of diabetic retinopathy, succinate levels increase in the retina, activating GPR91 and up-regulating VEGF signaling [137] which plays a central role in mediating microvascular and macrovascular pathology in diabetes.

An intriguing link between ageing and subtle changes in the production or distribution of Krebs cycle intermediates and ageing is suggested by the life-extending *Dropsophila* gene *indy*, which functions as an ion transporter with similarity to mammalian sodium–dicarboxylate cotransporters [138]. Mutations which impair this gene’s function may extend *Drosophila* lifespan by 80% and knockdown of the homologous citrate transporter ceNaC-2 increases the lifespan of nematodes [139].

One striking biochemical correlate of ageing with a clear connection to mitochondrial function is the decrease with age of nicotinamide adenine dinucleotide (NAD); in vivo NAD assay reveals the intracellular NAD contents and redox state in healthy human brain and their age dependences [140]. Structurally NAD comprises two nucleotides, adenine and nicotinamide, linked by their phosphate groups and can exist in reduced (NADH) and oxidized (NAD^+^) forms. NAD was first described as an essential cofactor in the fermentation of glucose by yeast extract by Sir Arthur Harden in 1906, occurs in all living cells and was initially thought to function mainly as a cofactor in transferring electrons in redox reactions. It is now appreciated that NAD is involved in a wide variety of regulatory activities and has intimate links both to mitochondrial function, organismal health and ageing. The direct link between NAD levels and ageing is indicated by the observation that NAD repletion enhances life span in mice [141].

NAD is produced via three different pathways, with initial starting points of nicotinic acid (Preiss–Handler pathway (PHP), tryptophan (the de novo synthesis pathway) and nicotinamide (NAM—the salvage pathway (see Figure 5). Within mitochondria, NAD functions as an electron acceptor at four steps of the TCA cycle and in fatty acid oxidation, in which reactions it is converted to NADH. The NADH then acts as an electron door at complex I of the electron transport chain. The mitochondrial and cytoplasmic NAD pools are distinct. NAD is made in the cytoplasm and shuttles into the mitochondrion via NAD/NADH shuttle, principally the malate-aspartate and glycerol-3-phosphate shuttles. However, it appears that mitochondria may have an independent NAD biosynthetic capability [142].

In addition to its role as an electron carrier in redox reactions, NAD has important functions in multiple pathways and mechanisms with relevance to ageing. It is a substrate for the DNA repair enzyme poly-ADP-ribose polymerase (PARP), which builds oligomers of ADP-ribose attachments to histones at sites of DNA damage [143]. PARP activation is related to health and longevity in a double-edged manner. Although DNA damage repair is a prosurvival activity, PARP is an avid consumer of both NAD and ATP [144] and depletion of these metabolites by PARP activation is a cause of cell death in some situations [145]. Having lower affinities for NAD than PARP, Sirt enzymes compete for the dwindling pool of NAD less efficiently than PARP and their activities, which include activation of mitochondrial biogenesis [146,147,148,149] are curtailed. The situation is exacerbated by PARP mediated inhibition of glycolysis via the inhibition of hexokinase 1 [150]. Under acute stress conditions, PARP inhibition may preserve NAD and ATP levels and allow cellular survival [151]. However, PARP activity chronically correlates with species-specific lifespan [152] and centenarians have higher PARP activity than a control group of subjects between the ages of 20 and 70 [153].

In addition to repairing strand breaks in DNA, PARP activity may also be required for the maintenance of telomeres, the repetitive DNA structures found at the ends of chromosomes [154,155,156,157]. Telomere shortening occurs through life, has been shown to be the cause of the replication exhaustion of populations such as fibroblasts (first described by Hayflick and Moorhead in 1961) [158] and has been proposed as a risk factor for age related diseases [159,160]. It is not clear whether declining NAD levels affect the activity of the telomerase sustaining PARP but if so, it would link the reduction in NAD levels with age to an important driver of cellular senescence.

CD38 (cluster of differentiation 38), also known as cyclic ADP ribose hydrolase is a multi-functional enzyme which, amongst other activities, catalyzes the cyclisation of NAD to produce cyclic ADP ribose, which it also uses as a substrate to produce linear ADP ribose [161]. Originally identified as a cell surface antigen present on cells of the immune system, it is now known to be expressed ubiquitously in all tissues. Its products cyclic ADP ribose (cADPR) and nicotinic acid adenine dinucleotide phosphate (NAADP) regulate calcium concentrations by acting at the ryanodine receptor (cADPR) or at acidic calcium stores (NAADP) [162]. CD38 is a major consumer of NAD [163], is induced under inflammatory conditions [164] and its expression increases with age [165,166]. As general systemic inflammation also increases with age [167,168,169,170,171] a causal relationship may exist between increased CD38 expression and decreasing NAD levels. The major age-related changes in NAD producing and consuming enzymes are indicated in Figure 6.

Accumulation of dysfunctional post-mitotic senescent cells is also a general feature of ageing and clearance of these alleviates some ageing associated disorders [172,173,174]. As CD38 is induced by factors secreted by senescent cells, this provides a further link between CD38, NAD and ageing [175]. The central role of CD38 is supported by the observation that a potent CD38 inhibitor ameliorates metabolic dysfunction in aged mice [176]. A direct link between these processes and mitochondrial dysfunction was demonstrated by Camacho-Pereira et al. [166] who showed that although expression levels of Sirt3 were similar in one year old wild type and CD38 knockout mice, the level of mitochondrial protein acetylation in the knockouts was lower than that of the wild type and the activity of liver Sirt3 in the presence of endogenous levels of liver NAD was 3.5 times higher in CD38KO than WT mice. Intrinsic Sirt3 activity was similar when measured in the presence of excess NAD, suggesting a causal relationship between the higher CD38, lower NAD and lower SirT3 activity observed with ageing.

### 5.2. Mitochondrial Proteases

A set of proteases mainly or exclusively expressed in mitochondria carry out a diverse array of functions. These include participating in mitochondrial proteostasis by carrying out various steps of the mitochondrial unfolded protein response, removing mitochondrial-targeting pre-sequences from cytoplasmically synthesized but mitochondrially destined proteins, rapid degradation of short-lived regulatory proteins and regulating mitochondrial dynamics, mitophagy and apoptosis pathways [177]. These proteases number approximately 20–25 depending on the criteria used to define them (e.g., exclusively mitochondrial vs expression in both cytoplasm and mitochondria) and have been reviewed by Quiros et al. [178]. There are in addition a few proteolytically inactive pseudoproteases which play important roles in spite of their lack of catalytic activity, such as the participation of UQCRC1 and UQCRC2 in the respiratory chain and the regulation of the activity of cognate proteases as in the case of PMPCA (a-MPP). A few clinical phenotypes are associated with mutations in some of these proteases; interestingly, the majority of these have a neurological presentation (e.g., Gilles de la Tourette syndrome resulting from a defect in Mitochondrial inner membrane protease 2 [179], Parkinson’s Disease with Ser protease HTRA2 [180] and PARL (Presenilins-associated rhomboid like) [181].

In addition to these clear clinical syndromes, changes in some mitoproteases have been linked to ageing [182]. In some cases, this may be guilt by association with processes that are known to be impaired with increasing age [183,184]. However, there are examples of a clear association between changes in expression or activity of a specific protease and aspects of ageing. The mitoproteases LONP has been associated with at least three functions including destruction of oxidized proteins, mtDNA metabolism and chaperone activities with the Hsp60–mtHsp70 complex. Homozygous deletion of LONP1 causes early embryonic lethality, suggesting that its functions are essential for life [185]. LONP mRNA and activity decrease during ageing [186] and the decrease in transcripts in muscles of aged mice could be partially reversed by caloric restriction, a procedure known to impede ageing in some experimental situations [187]. Conversely, its upregulation protects against oxidative stress [188]. Bezawork-Geleta et al. [189] showed that one mechanism of protection against protein aggregation is by the direct degradation of misfolded proteins independent of the mitochondrial unfolded protein response. Exercise, another intervention associated with prolonging longevity was, similar to caloric restriction, able to reverse the changes in LONP expression in skeletal muscle [190]. These data have been rationalized by suggesting that changes in the level of LONP are affected by both age and activity, the difference between heart and skeletal muscle being the requirement for ongoing activity in the former but not the latter [191].

The presenilin-associated rhomboid-like protein (PARL) is a key regulator of mitochondrial integrity and function. The mitochondrial kinase PINK1, a critical component in the control of mitophagy (discussed later), was the first identified substrate for PARL [192]. By cleaving PINK1 within its transmembrane domain, PARL prevents the accumulation of PINK1 on the mitochondrial membrane and results in its degradation by the ubiquitin–proteasome system [193]. In depolarized mitochondria, however, PINK1 insertion into the IM is impaired preventing cleavage by PARL, as a result of which it accumulates on the outer mitochondrial membrane and activates the ubiquitin ligase Parkin to induce mitophagy [194]. In addition to its role in regulating mitophagy, PARL regulates apoptosis via cleavage of the pro-apoptotic mitochondrial protein Smac (DIABLO); this is released into the cytosol after cleavage, where it inhibits *inhibitors of apoptosis* (IAPs) and loss of PARL impairs the induction of apoptosis. In addition to these roles controlling mitochondrial and cell fate, PARL may be a mediator of mitochondrial-nuclear signaling via its autocatalysis to generate a small peptide Pbeta [195]. PARL mRNA levels have been reported to decline with age [196]. These authors reported that PARL mRNA and mitochondrial mass were both reduced in elderly subjects and in subjects with type 2 diabetes mellitus. Muscle specific knockdown of PARL in mice induced lower mitochondrial content, decreased PGC1α protein levels and impaired insulin signaling, while suppression of PARL protein in healthy myotubes lowered mitochondrial mass and increased the production of reactive oxygen species.

Two proteases of the inner mitochondrial membrane, Oma1 and YME1L, are required for the processing of OPA1, a component of the inner mitochondrial membrane that is responsible for maintaining cristae and for inner membrane fusion during mitochondrial dynamics. OPA1 is processed during maturation by YME1L to a family of membrane-inserted long forms with multiple functions. Under stress conditions, Oma1 is activated and further processes these long OPA1 isoforms to short soluble forms, upon which cristae structure is lost and cytochrome C released to the cytoplasm, where it triggers the apoptotic cascade [197]. The intimate relationship between the activity of these two proteases and mitochondrial function and health is seen when the balanced mitochondrial fission and fusion events they maintain was disrupted by cardiac-specific ablation of YME1L [198]. This activated Oma1, which cleaved long OPA1, triggering mitochondrial fragmentation with an associated dilated cardiomyopathy and heart failure. Cardiac function and mitochondrial morphology were rescued by Oma1 deletion, which prevented OPA1 cleavage. Defects in mitochondrial dynamics have been associated with ageing [199] and it is tempting to assume that changes in the expression or activity of such key enzymes as YME1L and Oma1 (particularly as the latter is activated under a variety of stressful conditions) are likely to be involved, although no direct evidence for age-related changes have been reported as yet.

Genetically modified mice have also provided direct evidence for a role in ageing of other mitochondrial proteases, including Htra2, the loss of which in non-neuronal tissues causes premature ageing, owing to an increase in mtDNA deletions [200]. An inactivating mutation of mouse Immp2l induces an increase in oxidative stress in several organs [201]. The mutants display multiple aging-associated phenotypes, including wasting, sarcopenia, loss of subcutaneous fat, kyphosis and ataxia. The loss of IMMP2L also initiates a cellular senescence program [202]. Given their multiple roles in essentially all aspects of mitochondrial function and regulation, it will not be surprising if increasing evidence is found for a role in ageing of many mitochondrial proteases.

### 5.3. Cardiolipin

Cardiolipin is a non-bilayer forming phospholipid dimer with a wide diversity of molecular forms. It is expressed almost exclusively in mitochondria, where the vast majority is found on the inner mitochondrial membrane [203]. Its structure confers unique biophysical properties on the molecule [204] and it is both essential for life (knockout mice lacking the key biosynthetic enzyme PTPMT1 are embryonic lethal before day 8.5 [205]) and a key component of multiple mitochondrial activities. It interacts with membrane transport proteins including the phosphate carrier (PiC), pyruvate carrier, tricarboxylate carrier, the carnitine/acylcarnitine translocase and the essential ADP/ATP carrier (ANT) which allows the ATP formed by OXPHOS to be transferred from the IMM to intermembrane space [206,207]. It also interacts directly with complexes I, III and IV of the electron transfer chain, where it is required for the maintenance of the quaternary structure and function of the complexes [208,209]. It is also essential for the formation of higher order supercomplexes [210]. In addition to these roles, it is involved in several steps of mitochondrial dynamics and morphology, including fusion and fission [211,212,213] and mitophagy [214].

Its intimate association with the inner membrane sites of ROS generation renders cardiolipin particularly susceptible to oxidative damage in the form of lipid peroxidation [215]. Oxidation of cardiolipin impairs its function in bioenergetic processes [216,217]. Under stress conditions, cardiolipin may redistribute from the inner mitochondrial membrane to the outer membrane, where its exposure can trigger either mitophagy [218] or apoptosis [219,220]. Cardiolipin is a direct binding partner for the proapoptotic protein Beclin 1 [221]. It appears to be involved in multiple steps of the apoptotic pathway, as it also interacts directly with caspase 8 [222] which ultimately leads to oligomerization of Bax and Bak, resulting in OMM permeabilization and cytochrome C release.

Decreases in cardiolipin content with age have been reported in mitochondria of brain, liver and heart [223,224,225] and this is likely a contributor to age related decreases in energetic efficiency. In addition, as mentioned above, cardiolipin is a prime target for age related increases in oxidative damage, which could mediate increased age related maladaptive mitophagy and apoptosis. It has been a focus drug discovery efforts aimed at finding compounds able to protect cardiolipin and inhibit the energetic and cellular consequences of its depletion and oxidation [226].

## 6. Genetics of Mitochondrial DNA

Mitochondrial DNA is inherited in a non-Mendelian fashion and in mammals, with few and very limited exceptions, is inherited strictly through the oocyte (matrilineal inheritance), with active destruction of the male gamete mitochondrion shortly after fertilization [227]. In DNA inherited in this fashion (the human Y chromosome is another example), groups of alleles tend to remain associated for evolutionarily long times and such groups of alleles are termed haplotypes. Phylogenetically related groups of haplotypes are known as haplogroups, which are often associated with specific geographical regions, all of which have diverged during evolution from a hypothesized single ancestor, “mitochondrial Eve” [228]. Such haplogroups are characterized by sets of polymorphisms in mtDNA; we mentioned the ND3 encoding N haplogroup above. Other specific haplogroups have been associated with longevity and altered risk of specific disorders [229,230,231,232,233,234,235,236].

The effect of some of haplotypes on disease or longevity may be dependent on the context within which they are found. Thus, the J haplotype is associated with increased longevity in some specific populations] [237,238] and not others [239,240]. Intriguingly, it is also associated with an increased risk of some hereditary diseases such as LHON and optic neuritis in multiple sclerosis [241]. Far from being associated with an increase in mitochondrial function, the J haplotype shows lower VO (2max) [242], an indicator of reduced metabolic activity. The J2 (and U) haplotype also confers a reduced ROS production [243], which could be relevant to its association with longevity, although as noted below, the relationship between ROS production and longevity is not straightforward.

The observed dominance of uniparental inheritance of mtDNA raises the question why this mode of inheritance has evolved. The selective pressure may originate from the impaired fitness of heteroplasmic organisms, which in turn may be related to the need for interactions between different components of the electron transport chain, which have co-evolved for maximum complementarity and deviations from which are likely to impair functions. A mathematical model by Christie et al. [244] indicated that under all conditions of neutral or non-neutral mutation, selection against heteroplasmy could explain the evolution of uniparental inheritance. An additional consequence of a mammalian mtDNA bottleneck is that newly arising germ line mutations can increase to high levels within a generation [245].

The type of spontaneously arising mtDNA mutations, their rate of accumulation and the selective pressures operating on them, may differ between tissues that consist mainly of post-mitotic cells, such as brain and skeletal muscle and those with stem cell populations which continuously replenish lost cells, such as the GI tract and the immune system. Mitochondrial DNA is replicated independently of the cell cycle [246] and occurs in post mitotic cells but at a slower rate than that of dividing cells. In post-mitotic cells, mutations are thought more likely to arise from deletions due to the repair of oxidatively damaged mtDNA [247]. Reeve et al. [248] examined the frequency of both point mutations and deletions in brain substantia nigra neurons, which are highly susceptible to oxidative damage due to their dopamine metabolism and high iron content [249]. These cells displayed a high frequency of mtDNA deletion mutations but low levels of point mutations. In a direct comparison between skeletal muscle, heart and kidney, Liu et al. [250] detected a bias toward deletion mutations in skeletal muscle, with point mutation reaching higher levels in the heart and kidney. The distribution of mitochondrial mutations throughout the body is dependent on their origin; mutations transmitted via the oocyte are likely to be widely distributed throughout the tissue of the body, those arising in embryonic development or in stem cells of the adult may be restricted to specific niches, while those arising spontaneously in somatic cells comprise a large non-clonal set and any individual mutation may display a very restricted distribution.

An example of mutations newly arisen in stem cells populating their progeny was observed in the GI tract [251] where mtDNA mutations arising in stem cells of normal human gastric body units were seen to characterize all their differentiated descendants. A further example is the mitochondrial gene for the complex III component cytochrome b [43]. Five patients presented with exercise intolerance and displayed three different nonsense mutations (G15084A, G15168A and G15723A), one missense mutation (G14846A) and a 24-bp deletion (from nucleotide 15498 to 15521. There was no maternal inheritance of any of these alleles and they were confined to muscle tissue, suggesting that they arose during differentiation of the myogenic stem cell lineage.

The question of the extent and significance of mitochondrial DNA mutations arising spontaneously in somatic tissues, especially in post mitotic cells, is complex. Ma et al. [252] studied germline and somatic mutations in wild type and Polγ mutator mice. They found no detectable somatic mutations in wild type mice but the heteroplasmy of several different germline mutations increased with age, suggesting clonal expansion with ageing. Somatic mutations were detected in mice both homozygous and hemizygous for the Polγ mutation but as the authors note, only the homozygous mice display an accelerated ageing phenotype, making the relevance of the phenomenon unclear. However, the mouse in this respect is different from human, where age-related accumulations of somatic mutation has been observed [253,254]. Li et al. [255] examined skin fibroblasts from a 72 year old individual and found 34 mutations of which 14 occurred at heteroplasmy levels of >15%, while 6 occurred at levels of >80%, exceeding the 60% heteroplasmy that is considered the threshold for such mutations causing detectable deleterious effects on mitochondrial function. They modelled accumulation of somatic mutations based on human data and concluded that almost 90% of non-proliferating cells would be expected to have at least 100 mutations per cell by the age of 70 and almost no cells would have fewer than 10 mutations. Ye et al. [256] also documented the extensive occurrence of low frequency pathogenic mitochondrial mutations in healthy individuals.

In addition to emphasizing that there may be significant species differences in the accumulation of mtDNA mutations with age, these data suggest that the accumulation of spontaneously arising mitochondrial mutations in human are associated with ageing. Earlier theories on the selective pressures operating within mitochondrial DNA populations suggested a replication advantage of smaller molecules containing deletions but modelling this process and taking into account such factors as the much more rapid replication time for mtDNA (90 min) compared with the cell cycle time (10–24 h) indicated that random genetic drift could explain the increased proportion of mutated mtDNA with age in somatic tissues [257].

### Association of mtDNA Mutations with Disease and Ageing

We discussed the association with ageing of mutations in proteins encoded by the mitochondrial genome above. Here we turn to mutations in the mitochondrial genes encoding components of the mitochondrial translation machinery. The translation of mitochondrially encoded polypeptides occurs on mitochondrial ribosomes, which consist of 12S and 16S rRNAs and about 80 nuclearly encoded proteins [258]. Because mutations in the translational machinery can affect all proteins products of mitochondrial ribosomes, such mutations are associated with many pathologies. These usually manifest as deficits in the oxidative phosphorylation catalyzed by polypeptides translated on mitochondrial ribosomes [259].

All of the RNA components of the mitochondrial translation machinery are encoded in mtDNA, the protein components of the ribosome and accessory proteins such as aminoacyl tRNA synthetases being nuclearly encoded. Mutations in nuclear genes encoding protein components of the ribosome affect the accuracy of translation and are associated with a diverse range of pathologies including ovarian insufficiency [260], encephalopathy [261], Leigh syndrome [262] and stress related behavioral alterations in mice [263]. A similar diversity of presentation is seen in mutations affecting mitochondrial aminoacyl tRNA synthetases and the reasons for this diversity from an apparently similar mechanism are discussed by González-Serrano [264].

Mitochondrial tRNAs have several structural features that distinguish them from their cytoplasmic counterparts and mitochondria also deviate from the “universal” genetic code (reviewed in [265]. Mutations in mtRNA (or in enzymes related to their post transcriptional modification) are causative of a range of pathologies including myopathy [266,267], hearing loss [268,269] cardiopathy [270,271] and encephalopathy [272,273] and are associated with more than half of all mtDNA-associated human disease [274]. Aberrant post translational modification of mitochondrial tRNAs is also associated with disease. The A8344G mutation in mitochondrial tRNA^(Lys)^ is responsible for myoclonus epilepsy associated with ragged-red fibers (MERRF). In this mutation, the normally modified wobble base (a 2-thiouridine derivative) remains unmodified and this results in defective translation [275]. A mutation im.8344 A > G in tRNA^lys^ was shown to cause a defect in translation and stability of nascent polypeptide chains in MERRF (myoclonus epilepsy, ragged-red fibers) patients [276].

Although many pathologies were associated relatively early with mutations affecting tRNA, it has been more difficult to link mutations in the ribosomal RNAs with pathology. The disruptive potential of mutations in 16s rRNA was studied by Elson et al. [277] and in the 12S rRNA by Smith et al. [278] using a technique called heterologous inferential analysis (HIA), which integrates atomic resolution structural data with information on the conservation of overall structure of rRNAs. These studies revealed that mutations able to compromise function were indeed found in clinical conditions, of which the drug-induced hearing loss caused by A1555G and C1494T mutations at a highly conserved site of the 12S rRNA gene the best characterized [279,280]. Mutations in the 16S RNA are linked to myopathy [281] and cardiomyopathy [282].

Synergistic interactions between different mtDNA mutations within the same mitochondrion may affect ageing and function even when the individual mutations have no measurable effect on fitness. Reichart et al. [283] studied the effects of mtDNA mutations in a complex IV component and in the mttRNA^Arg^ gene on lifespan, learning and memory. Single point mutations in either gene had little impact on these parameters but mice with the combination of both mutations had a reduced life span and showed deficits in learning and memory. A high level of neocortical superoxide was seen in the double mutants compared with the single mutants. Simple increase in the copy number of mtDNA (achieved by manipulating TFAM expression) reduces the severity of pathology in a heteroplasmic mouse bearing a mutation in mt tRNA^Ala^ although the levels of heteroplasmy remained the same [284]. Conversely, a reduction in mtDNA levels worsened the phenotype in postmitotic tissues, such as heart, although enhanced clonal expansion and selective elimination of mutated mtDNA lead to a beneficial effect in rapidly proliferating tissues, such as colon.

The preceding survey is not exhaustive but should be sufficient to make the point that mutations in mitochondrial DNA, (whether in genes encoding subunits of the electron transfer chain or in mt tRNA or rRNA) are frequently associated with a decreased efficiency in ATP synthesis and increased generation of ROS, reflecting the specialized contributions of mitochondrially encoded components to the generation of ATP via electron transport. In addition, specific mutations usually have a noticeably clear clinical presentation. This latter observation is difficult to reconcile with the mitochondrial free radical hypothesis (see Section 6), under which accumulating mtDNA mutations leading to enhanced ROS production might be expected to affect all organs, with perhaps a more obvious manifestation in highly metabolically active tissues such as muscle, kidney and brain. In fact, specific mutations are associated with a relatively small number of often tissue specific diseases, including Mitochondrial Encephalopathy, Lactic acidosis and Stroke-like episodes. (MELAS), Leber’s hereditary optic neuropathy (LHON), Leigh syndrome, cardiomyopathy, myoclonic epilepsy with ragged red fibers (MERRF), deafness and myopathy. One observation bearing on this expectation is that at least some newly arising mutations that reach a significant degree of heteroplasmy are not randomly distributed across tissues. Using massively parallel sequencing techniques, Samuels et al. [285] assessed heteroplasmy across ten tissues and demonstrated that in unrelated individuals there are tissue-specific, recurrent mutations that were undetectable in other tissues of the same individuals. Though independent of each other, these all occurred in the regulatory region and the authors speculate that these are positively selected in a tissue specific manner. In addition, the effect of specific mitochondrial mutations can depend not only on the tissue but the state of differentiation; undifferentiated cybrids of the NT2 neuronal lineage cell line containing mitochondria bearing either the most common (11778) or the most severe (3460) LHON associated mutations did not differ from controls in ROS generation but upon differentiation displayed an increase in ROS production [286].

## 7. Mitochondrial Mutations and the Oxidative Damage Theory

An early hypothesis to explain cellular and organismal impairment with age linked the occurrence of mutations in mitochondrial DNA with the production of toxic free radicals. As indicated in Figure 7, mitochondria are the major source of free radical production in most cells, as a result of electron leak from complexes I and III of the electron transport chain, these electrons combining with molecular oxygen to produce the unstable reactive superoxide radical (•O^−^). This can directly attack molecules such a protein, lipids and nucleic acids and can give rise to various additional reactive species such as hydroxyl (•OH) and peroxynitrite ONOO^−^, not itself a free radical but a potent oxidizing species. There are a variety of biochemical defenses against free radical accumulation in cells, including antioxidant enzymes such as glutathione peroxidase, the linked enzyme systems of superoxide dismutase and catalase and a variety of free radical scavengers such as β carotene, vitamin C, vitamin E [287,288]. However, these defenses can be overwhelmed in situations where large amounts of free radicals are generated. This is seen when the catalase activity is insufficient to convert the hydrogen peroxide produced by SOD to water and oxygen, the excess hydrogen peroxide being reduced to reactive and toxic (•OH) by iron (Fe_2_^+^).

The Mitochondrial Free Radical Theory of Ageing (MFRTA) was proposed by Harman [289,290] and suggested that increasing accumulation of mutations with age in mtDNA led to defects in mitochondrially encoded components of the electron transport chain, with an associated increase in free radical production. These in turn produced further damage to mtDNA, leading to a self-propelled cycle resulting in eventual catastrophic damage accumulation. If anti-oxidant defenses are impaired with age, this might exacerbate the effect of increased ROS production. In the study of Reutzel et al. [44] reductions in brain catalase and SOD2 were seen at 18 months in mouse, suggesting that anti-oxidant responses may be impaired by this age. Such a reduction in anti-oxidant defenses with age was also seen in the rat as suggested by lowered expression of the antioxidant enzymes peroxiredoxin III (Prx III) and superoxide dismutase 2 (SOD2) [291]. In this study, levels of catalase decreased in most tissues but increased in muscle. These authors reported no correlation between the SOD, CAT and GSH-Px activities and the peroxidative status of the organs measured by malondialdehyde (MDA) content, which provides an important caveat in interpretations of changing level of expression of antioxidant enzymes.

Harman’s hypothesis elegantly linked two concepts (1) the known damaging effect of high levels of free radicals on organismal function and (2) the association of increasing accumulation of ROS-associated mtDNA mutations with age. However, many more recent observations have suggested difficulties in the original formulation of this theory. First, several observations suggested that long-lived species do not always demonstrate lower levels of ROS and the accompanying oxidative damage. Lewis et al. [292] compared the long-lived naked mole rat with mouse, a relatively short-lived rodent and found levels of ROS production and antioxidant defenses similar between the two species. The naked mole rats show high, steady state levels of oxidative damage even under unstressed conditions and still have an exceptionally long lifespan.

Second, loss of antioxidant defenses would be predicted to shorten lifespan but this was not found invariably to be the case. Loss of catalase, thought to be an important component of the endogenous anti-oxidant system, at least in *Caenorhabditis*, has no effect on lifespan [293,294]. and overexpression of catalase, either alone or in combination with superoxide dismutase did not increase lifespan [295]. It should be noted, however, that loss of either glutathione transferase or thioredoxin did reduce life expectancy in worms. Third, an increase in ROS signaling may lead to a paradoxical increase in life expectancy [296]. Yang and Hekimi [297,298] showed that an increase in measurable oxidative damage was without effect on the lifespan of a long lived strain of worms and Schulz et al. [299] showed that impaired glucose availability forced an increase in respiration and ROS generation in *Caenorhabditis* but this was accompanied by an increase, rather than a decrease, in lifespan.

The original form of the hypothesis in which oxidative damage caused by mitochondrial dysfunction initiates a vicious cycle of increased ROS production and increasing macromolecular damage culminating in manifestations of ageing and death is therefore untenable. Barja [300] argued that elements of the theory had been oversimplified and misunderstood and mounted a robust defense of the hypothesis, with a focus on low generation of endogenous damage and low sensitivity of membranes to oxidation in long-lived animals. More recently, he has incorporated these ideas into a unified theory of ageing [301]. We suggest that ROS are certainly linked to ageing, by mechanisms which may include directly damaging macromolecules (see our discussion of cardiolipin and supercomplexes) but that more subtle effects via epigenetic mechanisms may be of equal importance as discussed below. Figure 8 indicates the key elements of a revised MFRTA.

Fouquerel et al. [302] showed that selectively targeting the oxidatively modified base 8-oxoguanine to telomeres resulted in telomere shortening and genetic instability, directly linking a product of ROS generation with what is currently believed to be a key element in the ageing process [303]. Studies with two different progerioid mouse models have also indicated the possibility of more subtle effects of ROS production and accumulation. Ahlqvist et al. [304], working with the Polγ mutator mouse, showed that defects in neural and haemopoietic cells were present as early as embryogenesis. These defects manifested as a reduction of quiescent neural progenitors in the sub ventricular zone, a decreased ability to self-renew and abnormal differentiation of the HPCs. No direct evidence for oxidative damage to proteins, lipids or nucleic acids was found in the mutator mice but treatment with the antioxidant *N*-acetyl-l-cysteine could restore self-renewal in both populations. Similarly, their cardiomyopathy can be ameliorated by mitochondrially targeted overexpression of catalase [305]. These data suggest that small increases in ROS consequent on the mitochondrial mutations may control the ability of stem cell populations to renew and/or differentiate.

Pinto et al. [306] developed a mouse model in which double stranded breaks could be introduced ubiquitously in mtDNA by the mitochondrially targeted expression of an inducible endonuclease mito-ScaI, which leads to mtDNA depletion. In vitro, this was correlated with an upregulation of a transcriptional response associated with cell cycle arrest (increase in transcription of cell cycle arrest protein p21, Mdm2 (negative regulator of p53) and 14-3-3σ (cell cycle checkpoint controller). Only a small and statistically insignificant amount of H_2_O_2_ was seen 30 min after induction of the endonuclease (by which point mtDNA had already fallen to 40% of control levels), with larger amounts of H_2_O_2_ produced at 24 h. Treatment with *N*-acetyl-l-cysteine blocked the increases in cell cycle gene expression. Systemic induction of indmito-PstI for five days lead to an accelerated ageing, which became apparent at 6 months after induction. Although by 5 days no changes were seen in cell cycle arrest genes, at two days after induction Cdkn1a and 14-3-3σ transcripts levels were upregulated, as was p-MDM2, indicating stabilization of p53. These data indicated that cell cycle arrest signaling occurred before the accelerated aging phenotype. However, in the mitotic tissues in which the ageing phenotype was most obvious, no mtDNA depletion was found. These observations suggest a complexity in the ageing phenotype caused by these mitochondrial mutations, which is likely multifactorial. The reversal of some of their effects by antioxidant treatment suggests that ROS may indeed play a significant role but that this is related to aberrations in its normally physiological function rather than simply as a result of oxidative damage.

An indication of the need for subtlety in thinking about the relation of ROS to ageing and lifespan comes from the finding by Bazopoulou et al. [307] that a transient (10 h) increase in ROS in early development of *Caenorhabditis* triggers epigenetic mechanisms that result in increased stress resistance and redox homeostasis that in turn result in life span extension, the mediating mechanism being global decreases in histone H3K4me3 levels. A biphasic response such as this, where low levels of a potentially damaging molecule which is toxic at higher doses triggers the upregulation of cellular defense mechanisms that are ultimately beneficial to life expectancy was termed mitohormesis by Tapia [308].

## 8. Cellular Programs and Aging–Mitochondria in Stem Cells and Senescence

Stem cell populations in adult tissues constitute a pool of committed progenitors for tissue renewal. These populations are typically long lived and quiescent and undergo self-renewal by proliferation and differentiation to specialized tissue cell types. An appropriate balance between these two fates is required for tissue homeostasis. Stem cell function is impaired with age and is responsible for many of the obvious hallmarks of normal ageing such as sarcopenia, decreased immune function and slower wound healing [309].

Mitochondria have been shown to play a role in determining the fate of stem cells by mechanisms including energetics [310], changes in mitochondrial dynamics [311] or the mitochondrial unfolded protein response [312]. The role of Kreb’s cycle intermediates as metabolic regulators has already been discussed and such regulation also impinges on stem cell fate decisions [313]. The requirement for a shift from the typical stem cell glycolytic phenotype to an increased reliance on OXPHOS with differentiation was mentioned above. A declining ability of mitochondria to carry out efficient OXPHOS or quality control through dynamics and the mtUPR may therefore have an impact on one key aspect of ageing, the inability effectively to replenish stem cell pools and produce specialized post mitotic cell types in various organ systems. Katajisto et al. [52] followed the fates of old and young mitochondria during the division of human mammary stem like cells and found that such cells apportion aged mitochondria asymmetrically between daughter cells. Daughter cells that received fewer old mitochondria maintained stem cell traits and inhibition of mitochondrial fission disrupted the age-dependent subcellular segregation of mitochondria and caused loss of stem cell properties in the progeny cells.

ROS have been recognized as an important physiological signaling trigger. Physiological ROS signaling is mediated by reversible changes in the oxidation status of sulphur-containing amino acids cysteine and methionine which function as redox sensitive molecular switches. The delicate balance between ROS signaling and stem cell function has been reviewed by Tan and Suda [314] and Zhang et al. [315] and as discussed in the preceding section, it appears that relatively subtle changes in the level of ROS production due to accumulating mitochondrial mutations may have a much more dramatic impact through this mechanism than through accumulation of gross damage to macromolecules.

In addition to loss of stem cell self-renewal and potency, another aspect of ageing is the accumulation in many tissues of populations of post-mitotic and non-functional senescent cells. These cells may play a beneficial role in some aspects of development and tissue remodeling after injury and their phenotype has been seen as an escape route from oncogenic transformation [316]. Their accumulation is generally a response to some level of stress, either chronically as a response to telomere shortening or more acutely as a result of a variety of stressors including radiation damage, oncogene expression, disturbed proteostasis or oxidative stress. These stressors induce DNA damage response which activates a set of effector programs leading to senescence induction and maintenance, the two best studied centering on p53 [317] and p16INK4A ] [318]. In spite of their possibly beneficial roles in some specific circumstances, the accumulation of senescent cells with age has a deleterious effect on organ function in a range of tissues. This is partly due to their loss of differentiated function, as they contribute to tissue mass without performing tissue specific roles. In addition, they release a complex pro-inflammatory mixture of cytokines, growth factors and enzymes (the senescence associated secretory phenotype, SASP) that can drive bystander cells into senescence and contribute to the damaging inflammation that accompanies ageing [319,320,321,322,323,324]—see Figure 9.

Senescent cells are themselves a source of ROS as a result of their dysfunctional mitochondria [325] and ROS-dependent feedback loops can positively reinforce the senescence pathway [325,326]. One trigger of the DNA damage response and induction of senescence is the telomere shortening mentioned above. As noted, accumulation of oxidized bases in telomeric DNA can result in accelerated telomere shortening [302], providing a link between the increased ROS production of dysfunctional mitochondria with a key driver of the ageing process.

Defective mitochondria are also linked to the induction of senescence via their role in triggering the SASP. Mitochondrial DNA is normally absent from the cytoplasm of healthy cells, as defective mitochondria are dealt with by mitophagy, in which the entire organelle is enclosed by a phagocytic membrane and the contents are degraded. This prevents exposure of the cytoplasm to mtDNA. However, under conditions of cellular stress or when mitophagic mechanisms are insufficient, damaged mitochondria can release DNA, which is sensed as a damage associated molecular signal by the intracellular enzyme cGAS [327]. This in turn signals via the gene regulator STING to activate IRF3 and NFκB regulated proinflammatory genes. This mix is able to induce senescence in bystander cells. Induction of mitochondrial DNA release in response to interleukin1, a cytokine commonly induced early in inflammatory conditions, was recently described [327], implying the existence of positive feedback loops between inflammation and the induction of the SASP. Increased levels of mitochondrial DNA have been found in body fluids of a number of autoimmune pathologies, including rheumatoid arthritis [328] and Lupus [235].

A general increase in circulating mtDNA after the fifth decade was reported by Pinti et al. [329]. Although Jylhävä et al. [330] did not find an association between circulating mtDNA and age, they did find that levels of mitochondrial DNA were correlated with increasing frailty.

## 9. Mitochondrial Quality Control

### 9.1. Mitophagy

Maintenance of the function and structural integrity of mitochondria is tightly coordinated by quality control mechanisms. Balanced control of mitochondrial dynamics through the processes of fission and fusion, in addition to regulation of the selective removal of damaged mitochondria by mitophagy make up the key pathways of mitochondrial quality control as shown in Figure 10. Their importance is indicated by the fact that impairment of any of them is associated with poor cell function and viability, which is manifested in several human pathologies including age-related diseases. Given that these processes operate in a time scale of minutes to hours, in the next part of this review we will discuss why mitochondrial quality control declines with age and fails to prevent lifetime deterioration in mitochondrial function.

Mitophagy is an evolutionary conserved pathway and primarily serves a housekeeping role by recycling mitochondria and adjusting the pool to respond to cellular needs. It can also act as stress response pathway that selectively marks and eliminates damaged mitochondria in order to maintain a healthy mitochondrial population [331]. Furthermore, mitophagy is also responsible for elimination of mitochondria during red blood cell differentiation, degradation of sperm-derived paternal mitochondria following fertilization, maturation of T-lymphocytes and differentiation of retina ganglion cells [332,333,334,335,336]. The involvement of mitophagy in all these processes underlines its importance in cellular homeostasis and highlights its role in preserving cellular function and organismal health span.

Over the past decade, different pathways regulating mitophagy have been identified. The best characterized is mediated by the kinase PINK1 (PTEN-induced kinase 1) and the E3 ubiquitin ligase Parkin, mutations in both of which have been associated with autosomal recessive forms of Parkinson’s disease [337,338]. Under steady state conditions and in healthy mitochondria PINK1 is rapidly turned over via a sophisticated import mechanism, which is dependent on the presence of an amphipathic mitochondrial targeting sequence and membrane potential. Both of these elements are required for PINK1 to cross the outer and inner mitochondrial membranes, through the TOM and TIM complex respectively and subsequent cleavage by the proteases MMP and PARL [339]. Proteolytic processing of PINK1 by PARL, exposes a phenylalanine at amino acid 104 that leads to the retrotranslocation of the protein to the cytosol and allows it to be recognized by N-end rule E3 enzymes which leads to its constitutive degradation by the proteasomal pathway [340].

Conditions that disrupt mitochondrial function and lead to damage, such as loss of membrane potential or accumulation of protein aggregates in mitochondria, warrant and elicit a mitophagic response through the stabilization of PINK1 on the mitochondrial outer membrane [341]. Under stress conditions, physiological import and degradation of PINK1 in mitochondria is blocked which leads to its accumulation on the outer mitochondrial membrane where it phosphorylates ubiquitinated substrates and the E3 ubiquitin ligase Parkin [342,343,344,345,346]. Both of these phosphorylation events trigger activation and recruitment of Parkin to the mitochondria where it drives the ubiquitination of outer mitochondrial resident proteins leading to a positive feed forward mechanism by generating additional substrates for PINK1 [347]. The local increase in ubiquitin on mitochondria leads to the recruitment of the autophagic machinery to engulf selectively only the damaged mitochondria [348,349]. As such, this mechanism ensures a tight regulation of mitophagy and safeguards the healthy mitochondrial pool from being eliminated.

Several publications provide evidence indicating that aging tissues display an accelerated decline in mitophagy efficiency and an aberrant accumulation of damaged mitochondria that affect health and lifespan in different organisms. Two recently developed tools to monitor mitophagy in vivo have been instrumental in uncovering the mitophagic state during aging. Using the fluorescent mitochondrial-matrix targeted mitophagy reporter mt-Keima in mice, the group of Toren Finkel [350] observed marked variations in the basal levels of mitophagy within and between tissue types. Low rates of basal mitophagy were observed in the thymus but in contrast, cells with high-energy demands such as hepatic and kidney cells displayed high rates. Similarly, tissues with postmitotic cells, like the heart and brain, displayed enhanced levels of mitophagy, highlighting the need for fine-tuned mitochondrial function to sustain their homeostasis. Another interesting observation the group made was that within the brain there was significant heterogeneity in the basal levels of mitophagy. Areas enriched with neural stem cells such as the dentate gyrus and the lateral ventricle displayed marked enhancement in the levels of mitophagy. More importantly, a 70% reduction in mitochondrial clearance was observed in the dentate gyrus of 21-month old mice compared to 3-month old mice, providing evidence that mitophagy rates decrease during aging and may underlie the decline in memory and learning during aging as well. Finally, a similar decline was observed in the Huntington’s genetic model of neurodegeneration, further highlighting the importance of mitochondrial clearance in pathological conditions that are commonly manifested in aged populations.

Similar observations were reported using another mouse model to monitor mitophagy in vivo by the group of Ian Ganley [351]. In this study, the fluorescent reporter is called mitoQC and while both reporters rely on pH changes to monitor mitophagy, mitoQC differs from mtKeima in that it bypasses the overlap seen between the emission and excitation spectra. The mitoQC reporter localizes in the outer mitochondrial membrane and as such it can misreport outer membrane turnover for mitophagy changes. Nonetheless, this study corroborated very similarly the results of the mtKeima mouse and provided further evidence for the spatially restricted nature of mitophagy as a process within distinct cellular subtypes. Pronounced mitochondrial turnover was observed in kidneys and more particularly enhanced mitophagy was shown in the tubules of the renal cortex. Given that the renal system regulates vital functions such as water homeostasis, acid-base and electrolytic balance and arterial pressure, this study indicates the nephron as a promising target for therapeutic intervention in the context of mitochondrial turnover. Furthermore, this study reported increased rates of mitochondrial clearance in other tissues of high metabolic demand such as the brain and skeletal muscle. The most pronounced effects were seen in the Purkinje cells of the cerebellum and muscle fibers of the tongue.

In a follow up study [352], the same group reported that basal mitophagy occurs independently of PINK1 suggesting that other pathways may play a more prominent role under steady state turnover, while the PINK1/Parkin pathway may be activated in situations of cellular stress. In that respect, crossing Parkin KO mice with the mtDNA-mutator mouse model that carries a knock-in mutation that compromises the proofreading ability of Polγ and mimics the signs of aging, partially rescued mitochondrial dysfunction and preserved dopaminergic substantia nigra neurons suggesting that this pathway, while it may not significantly impact basal levels of mitophagy, is nevertheless critical for age-related pathological situations [353]. Another controversial issue with the PINK1/Parkin mitophagy pathway is that germline deletion has relatively subtle effects in mice. However, overexpression of Parkin or PINK1 in *Drosophila* melanogaster leads to lifespan extension [354,355]. Furthermore, a recent study in rhesus monkeys that deleted PINK1 by CRISPR/Cas9 showed by MRI and EM significantly decreased gray matter and degenerated neurons in the cortex, substantia nigra and striatum in adult monkeys [356,357]. This suggests that there are species specific differences and studies in mice may not necessarily capture fully the significance of the PINK1/Parkin mitophagic pathway during aging.

Substantial data supporting a role of declining mitophagy in tissues that contribute to aging phenotypes has been generated over the past decade. In addition to the contribution of dysfunctional mitochondria to neurodegeneration, mitochondria with decreased membrane potential was also observed in aged skeletal muscle satellite cells isolated from humans or mice, as well as increased mitochondrial accumulation inside autophagosomes indicative of inefficient mitophagy [358,359]. Furthermore, decreased expression of autophagy and mitophagy genes including Parkin, was reported in the skeletal muscle of elderly women, which may underlie the low muscle mass and poor physical function [360]. Studies in *C. elegans* and *D. melanogaster* have also strengthened the connection of mitophagy to health and lifespan. For example, studies in *Drosophila* have shown that besides overexpression of mitophagy genes contributing to lifespan, overexpression of the mitochondrial fission protein Drp1, which is required for efficient mitophagy, also leads to prolonged health span [361]. Similarly, studies in the nematode have shown that genetic perturbations in the mitophagy pathway compromise longevity under conditions of stress, while iron depletion increases mitophagy and is required for longevity [362,363].

Multiple age-related pathologies including neurodegeneration, inflammatory diseases and myopathies are characterized by the accumulation of dysfunctional mitochondria and as such, there is extensive effort in the scientific community to identify therapeutic interventions that boost the selective removal of defective mitochondria. Naturally occurring compounds including spermidine and urolithin A have been shown to preserve mitochondrial function and cellular homeostasis via mitophagy enhancement. Spermidine supplementation in multiple model organisms including yeast, flies and nematodes promoted lifespan extension and more importantly ameliorated the hypertrophy, impaired diastolic function and arterial stiffness seen in mouse models of cardiovascular pathologies [364,365]. The general importance of clearing aged and defective molecules and organelles is also indicated by the observation that caloric restriction and resveratrol promote longevity through the Sirtuin-1-dependent induction of autophagy [366]. Similarly, urolithin A, which is a metabolite from pomegranate seeds, was very elegantly shown by the lab of Johan Auwerx to boost mitophagy, extend the lifespan of worms and improve muscle function and exercise capacity in old mice [367]. Very recently, urolithin A was used in a first-in-human clinical trial and administered to healthy, sedentary elderly individuals. Encouragingly, not only did the compound had a favorable safety profile but also induced a molecular signature of improved mitochondrial and cellular health, shown by upregulation of mitochondrial genes in skeletal muscle [368]. Given that 25% of the global population is projected to be older than 65 years by the year 2100 [369], finding therapeutic interventions that will boost health span is of paramount importance. In this respect, approaches that stimulate mitophagy present themselves as attractive therapeutic opportunities that could have widespread beneficial effects in preventing age-related decline.

### 9.2. Fission/Fusion

As discussed previously, mitochondria are not formed and do not exist as isolated entities but constitute an interconnected tubular network to which new molecular components are added and from which aged and damaged molecules removed by processes of biogenesis and mitophagy respectively. This network is subject to constant remodeling by processes involving fission and fusion and this remodeling serves to distribute newly synthesized components throughout the network and to segregate damaged components. Many studies have indicated that these highly regulated processes of fission and fusion, (generally termed mitochondrial dynamics) are essential for the health of the cell and their disruption experimentally has profound consequences, while altered morphology of the network towards either a highly fragmented or a hyperfused morphology is associated with disease and ageing. In addition, there are mitochondrial morphologies that do not fit this simple hyperfused/hyper fragmented axis, which nevertheless arise from disturbances to the fission/fusion process and which are associated with cellular dysfunction, such as the granular phenotype described by Hengst et al. [370]. We will briefly review the major molecular components of the mitochondrial dynamics system to provide context for the discussion that follows.

The dynamin related protein Drp1 plays an essential role in executing mitochondrial fission. Drp1 exists as a cytosolic protein which can form homo-oligomers and assemble on the outer mitochondrial membrane on phosphorylation of Ser616. Several molecules on the outer membrane can serve as receptors for Drp1, the major identified being mitochondrial fission factor (MFF), fission protein-1 (FIS1), mitochondrial dynamics protein-49 (MiD49) and mitochondrial dynamics protein-51 (MiD51) [371,372]. Drp1 forms an oligomeric ring around the mitochondrion at the point of fission and this undergoes a GTP hydrolysis driven conformational change, which constricts the mitochondrion and results in fission. This outer membrane mechanism is well characterized but additional proteins in the inter membrane space contribute to the process by assisting with inner membrane constriction [373].

Mitochondrial fusion is mediated by other dynamin related proteins, the best characterized of which are Optic Atrophy 1 (OPA1) on the inner membrane and Mitofusins 1 and 2 (Mfn1, 2) on the outer membrane [374]. OPA1 exists as multiple isoforms and long OPA1 is essential for maintaining the integrity of cristae. Under conditions of stress, the inner membrane protease Oma1 becomes activated and cleaves OPA1 to lower molecular weight isoforms which no longer function to maintain inner membrane structure and this or loss of either OPA1 or the mitofusins, results in hyper fragmentation of the network.

Age related changes in both the components of the dynamics machinery and in the balance between fission and fusion have been described in numerous organisms, including *Saccharomyces*, *Drosophila*, *Caenorhabditis* and mammals but a lack of consistency between tissues, species and functional consequences makes it difficult to draw generally valid conclusions. The mitochondria of middle aged *Drosophila* flight muscles are more elongated than those of younger flies and a transient induction of Drp1 for 7 days restored the morphology to that typical of younger ages, improved mitochondrial respiratory function, reduced mitochondrial (ROS) production and extended lifespan [361]. These effects were linked to improved proteostasis, emphasizing that changes in mitochondrial dynamics do not exert effects in isolation from other aspects of the quality control system. The bulk of the evidence in lower organisms tends to suggest that it is the balance between fission and fusion that is most critical to health and longevity and driving either alone is not conducive to longevity. Thus in *C. elegans*, pathologies associated with inhibition of one process can be reversed by simultaneous inhibition of the other [375]. However, in *Saccharomyces cerevisiae*, although double mutants lacking the genes dnm1 and deficient for both fission and fusion appear to have normal wild type mitochondrial morphology, they have a shorter lifespan, suggesting that the adynamic condition itself is maladaptive. This phenotype was particularly marked under conditions of nutrient stress, suggesting impaired ability to adjust mitochondrial metabolism to prevailing conditions [376].

Drp1 expression in mouse synaptosomal mitochondria was found to rise between 5 and 12 months of age, thereafter, declining to 24 months, while OPA1 and Mfn 1/2 showed the opposite pattern increasing after 12 months [25]. This suggests a shift to a pro-fusion state with age in these long-lived cells. Similarly, in mouse muscle, Leduc-Gaudet et al. [377] also found an age related increase in mitochondrial length and branching; although no major changes were noted in the expression of several proteins involved in mediating mitochondrial dynamics, they did note a change in the ratio between Mfn2 and Drp1 consistent with the more fused morphology. On the other hand, excessive fission is also associated with the pathology of several age related disorders.

As discussed by Loson et al. [371] the interaction between Drp1 and its receptors is critical for mitochondrial fission. Excessive fission resulting from the interaction of Drp1with Fis1 has been implicated in the pathology of several disorders, including Alzheimer’s Disease [378], septic cardiomyopathy] [379], amyotrophic later sclerosis [380] and Huntington’s Disease [381]. Lack of Fis1 in either nematodes or mammalian cells is associated with only mild fission defects but aberrant morphology [375]. In a model of diabetic nephropathy, overexpression of MAP kinase Phosphatase 1 was associated with enhanced glucose control, sustained renal function, attenuated kidney oxidative stress and inhibition of the renal inflammation response [382]. This overexpression also reversed the consequences of the mitochondrial hyperfragmentation seen during hyperglycemia, (decreased mitochondrial potential, elevated mitochondrial ROS production, increased pro-apoptotic factor leakage, augmented mPTP opening and activated caspase-9 apoptotic pathway). These effects were mediated by inhibition of a JNK-CaMKII-Fis1 pathway [382]. Zhang et al. [383] generated conditional knock-out Fis1 mice to allow for specific Fis1 deletion in adult skeletal muscle. Loss of Fis1 was associated with mitochondrial hyperfusion, respiratory chain deficiency and increased mitophagy, which was exacerbated under exhaustive endurance exercise. Sirtuin 3 inhibition induces mitochondrial stress in tongue cancer by targeting mitochondrial fission and the JNK-Fis1 biological axis [384] This suggests the importance of Fis1 mediated fission for maintenance of mitophagy both at rest and under stress. A role for Fis1 in non-Parkin mediated mitophagy initiated by the SNARE protein Syntaxin 17 (STX17) was indicated by the abnormal accumulation of STX17 on the mitochondrial outer membrane, followed by its recruitment of core autophagy proteins to form a mitophagosome [385].

This is an appropriate point to note that mitochondrial fission and fusion is a highly regulated process at the post-transcriptional level and this provides another way in which age-related changes in dynamics may occur in the absence of obvious changes in the transcriptional profiles of the major components of the dynamics machinery. As an example, the RNA binding protein Pumilio2 (PUM2) was shown by the group of Johan Auwerx [386] to be upregulated on ageing in *C. elegans*, where it inhibits translation of mRNA encoding Mff, impairing fission and mitophagy.

One possible consequence of these age-related changes is a reduction in mitophagy (poorly executed in larger mitochondria) and increase in the burden of oxidized and malfunctional proteins. Consistent with this, Navratil et al. [387] found that giant mitochondria accumulating in cultured rat myoblasts had low inner membrane potentials and did not fuse with each other or with normal mitochondria. Tezze et al. [388] reported an age-related decline in Opa1 levels in sedentary but not active human muscle, which they associated with age related sarcopenia. Opa1 deletion from mouse muscle led to a precocious senescent phenotype and premature death, via induced ER stress leading to muscle loss and systemic ageing. However, a similar elongation of mitochondria in senescent human endothelial cells, associated with a decreased expression of the fission-mediating Drp1 and Fis1, was interpreted as an adaptive response to excessive ROS production, which triggered mitochondrial fragmentation and loss of membrane potential in young cells but from which senescent cells were protected [389]. Of the two mitofusins, Mfn2 is the most prominently expressed isoform in skeletal muscle.

Sebastian et al. [390] showed that Mfn2 is expressed in mouse muscle at progressively lower levels with age and this appears to trigger an inhibition of mitophagy and accumulation of damaged mitochondria; genetic ablation of the gene in muscle generated a gene signature associated with age. Building on earlier work in which cardiac ablation of either Drp1 or Mfn1/Mfn2 produced extensive mortality by six weeks of age, Song et al. [391] showed that simultaneous knockout of all three proteins mitigated this pathology to a less aggressive cardiac hypertrophy. In this study, they also noted that cardiac Drp1 overexpression caused mitochondrial fragmentation but this was associated with neither mitochondrial dysfunction nor overt cardiac pathology to 93 weeks of age. These observations support the conclusion that an inability to perform dynamic fission/fusion is itself maladaptive and also provide an important caveat to assumptions that fragmented mitochondrial networks are necessarily dysfunctional.

Changes in mitochondrial functions such as ATP synthesis, ROS generation and membrane potential have been associated with morphologies deviating from the norm but the relationship between form and function and by implication, between the balance of fission and fusion, is context specific and resists expression as a simple linear correlation. Mitochondrial dynamic processes play a role not only in mitochondrial quality control but are increasingly appreciated as regulating the adaptive responses of mitochondria to nutrient status [392] which makes a connection between lifespan extending interventions such as caloric restriction and mitochondrial dynamics. In addition, the cycle of mitochondrial fission and fusion distributes mitochondrial DNA throughout the network and changes in the balance between the two processes can affect the segregation of mutant and wild type mt DNA. Malena et al. [393] studied the effect of silencing Drp1 or hFis1, both required for mitochondrial fission, in a cybrid heteroplasmic for the pathological A3243G mutation. Down regulation of either of these gene products was associated with increased levels of mutant mitochondrial DNA, suggesting a link between the accumulation of dysfunctional mtDNA mutations with impaired dynamics during ageing.

An obvious effect of the dynamic remodeling of the network is the redistribution of mitochondrial components, including lipids [394], proteins [395] and mitochondrial DNA [396]. However, the fission process does not necessarily generate two equivalent and homogenous daughter mitochondria. Twig et al. [348] showed that one daughter maintained high membrane potential and had an increased propensity for subsequent fusion events, while the other exhibited lower membrane potential with lower levels of OPA1 and a reduced potential for further fusion. Inhibition of fission by a dominant negative Drp1 decreased autophagy and led to an accumulation of oxidized proteins in mitochondria with a reduced respiratory capacity. These data suggest that uneven fission may be part of a mechanism for segregating damaged proteins differentially between daughter mitochondria whose subsequent fates (further fusion or mitophagy) result in selective removal of damaged mitochondrial components. A similar conclusion was drawn by Abeliovich et al. [349] on the basis of combined genetic and proteomic data in yeast. Proteins with different rates of mitophagic degradation were shown to be unevenly distributed throughout the mitochondrial network and this segregation was attenuated by inhibiting mitochondrial fission in a DNM1 (Drp1) deletion mutant.

Mouli et al. [397] presented a detailed model in which coordinated fission and fusion events accelerate the removal of damaged mitochondrial components by autophagy. Furthermore, the model predicts the existence of an optimal frequency of fusion and fission events that can maintain respiratory function at steady-state levels in the face of ongoing oxidative macromolecular damage. The implication is that any disturbance of this balance in either direction may result in the accumulation of damaged and dysfunctional mitochondria. These data emphasize that mitochondrial dynamic remodeling functions in combination with mitophagy to maintain optimal mitochondrial function. Terman et al. [398] argued that the accumulation of enlarged defective mitochondria as a result of inefficient dynamic mechanisms has particularly pronounced deleterious effects in long lived cells compared with renewing mitotic populations.

As originally noted, there is incomplete concordance between studies on the relationship between ageing and the mitochondrial dynamics system between different organisms and some effects described may be selective for the tissue under study. The most reliable general conclusion is that the ability to remodel the mitochondrial network rapidly in response to nutrient or stress conditions is essential for optimal mitochondrial function. These processes serve to redistribute newly synthesized molecules throughout the network, segregate dysfunctional mitochondria for recycling and contain the spread of deleterious mtDNA mutations. They are perturbed during ageing and it is the ability to engage in them, rather than the maintenance of a static mitochondrial network of any specific type that is most critical to cellular health.

## 10. Conclusions

A remarkable level of integration has resulted from 1.45 billion years of co-evolution since the original association between the proto-mitochondrial prokaryote and its proto-eukaryote host. It is probably the case that no aspect of cellular function is without an intimate connection with the mitochondrion and this is particularly evident when considering cellular responses to nutrient conditions, stress, disease and ageing. It is clear from the above discussion that declining mitochondrial function at all levels, from the genetic to the biochemical, accompanies the phenomenon of cellular ageing, which in turn underlies the progressive loss of organismal fitness with age. A key question is whether specific components of mitochondrial dysfunction exist in a causal relationship to age related loss of cellular fitness and if so, the directionality of this relationship. The free radical theory of mitochondrial ageing was the most comprehensive attempt to attribute such a causal role to the mitochondrion but its original formulation has lost support as the complex effects of ROS on ageing have become better appreciated. Similarly, the finding that a deleterious mitochondrial mutation must achieve approximately 60% heteroplasmy has made simple genetic models of mutation untenable. Several additional processes have been identified which may link mitochondrial biology with ageing. These include epigenetic rather than directly damaging effects of ROS, possibly linking ROS to telomere shortening, which is one of the best substantiated biochemical processes controlling cellular lifespan. Effects of mitochondrial function on cellular properties such as senescence and stemness are also likely to provide meaningful links, while a decline in the efficacy of the dynamic mechanisms regulating mitochondrial quality have also been implicated. The complexity of these processes and their inter relationships are still not fully understood and at this point it seems unlikely that a single linear cause and effect relationship between any specific aspect of mitochondrial biology and ageing can be established in either direction. It is probably more useful to think in terms of an interlocking web of mutually reinforcing relationships and in this sense indeed, mitochondrial function and ageing are intimate relations.

## Figures and Tables

**Figure 1 ijms-21-07580-f001:**
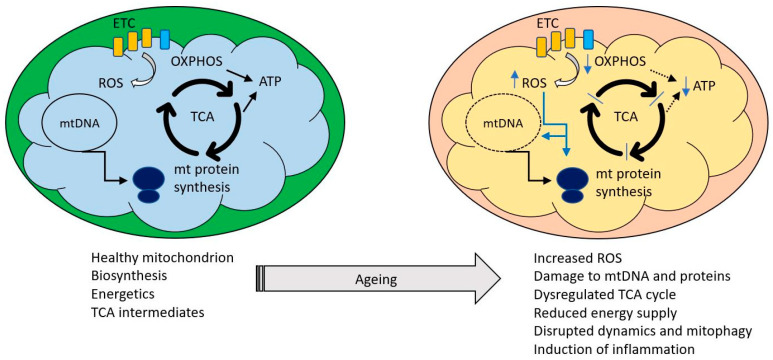
Healthy mitochondria perform a variety of functions including ATP production and synthesis of a variety of metabolites required for cell maintenance. Major processes are indicated by black arrows, blue arrows indicate changes occurring during ageing. Ageing is associated with an increasing impairment of these functions, manifested as excessive reactive oxygen species (ROS) production, oxidative damage to mitochondrial components, reduced ability to supply ATP, dysregulation of the tricarboxylic acid cycle and an increase in mitochondrially driven inflammation.

**Figure 2 ijms-21-07580-f002:**
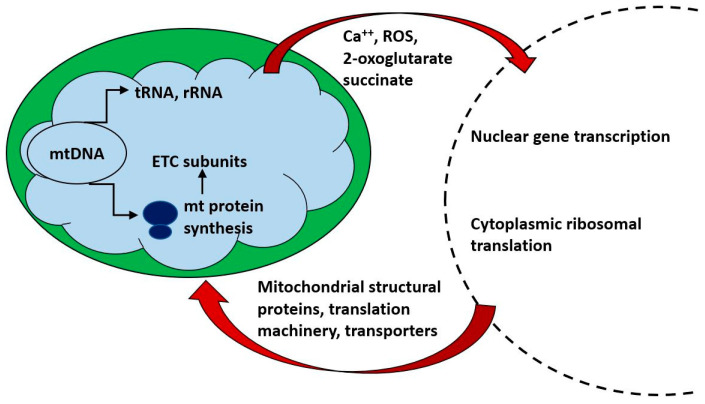
Mitochondria communicate bidirectionally with the nucleus and cytoplasm. Nuclear gene transcription is required for the synthesis of the vast majority of mitochondrial components, is tightly coordinated with mitochondrial protein synthesis and is responsive to the energetic requirements of the cell. Small metabolites (e.g., 2-oxoglutarate and succinate) act as ligands for G protein coupled receptors which may in turn affect cellular metabolism and nuclear gene transcription.

**Figure 3 ijms-21-07580-f003:**
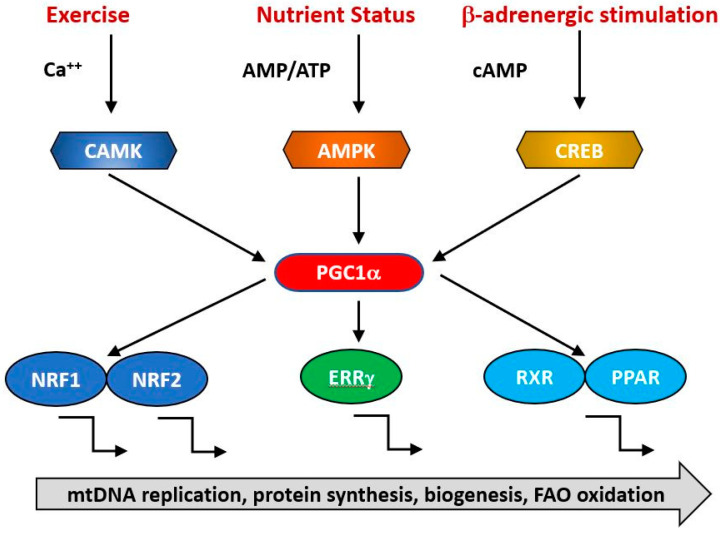
PGC1α is a transcriptional co-factor that regulates the transcription of nuclearly encoded mitochondrial proteins by interacting with transcription factors such as Nrf1/2, ERRg and the PPAR family. It acts as a hub to integrate signals from several pathways that monitor the energetic and nutritional status of the cell, including AMP-activated protein kinase (AMPK), calcium/calmodulin-dependent protein kinase (*CAMK*) and cAMP-response element binding protein (CREB).

**Figure 4 ijms-21-07580-f004:**
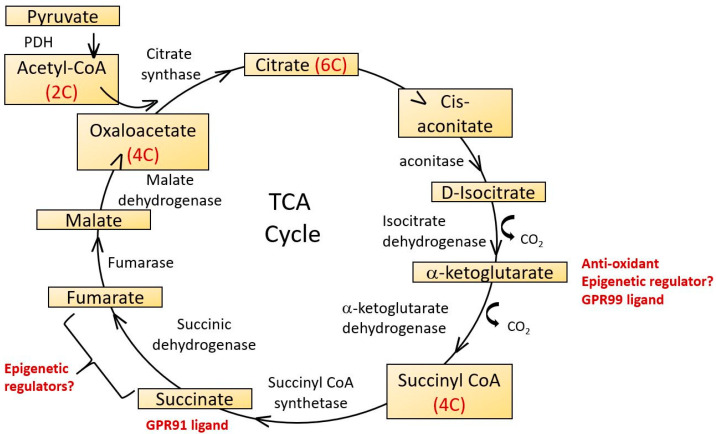
The tricarboxylic acid cycle (TCA), also known as the citric acid cycle of Krebs cycle occurs in the mitochondrial matrix. It functions to supply ATP, NADH and biochemical intermediates for a variety of metabolic pathways. 2-carbon acetyl CoA derived from glycolytic pathways reacts with 4-carbon oxaloacetate in the first step and a linked series of reactions results in the synthesis of two carbon dioxide molecules, three NADH, one FADH2 and one ATP molecule per cycle, regenerating oxaloacetate as it does so. Several TCA cycle intermediates have potential roles as epigenetic regulators and ligands for G protein coupled receptors. 2-Oxoglutarate is also known as α-ketoglutarate.

**Figure 5 ijms-21-07580-f005:**
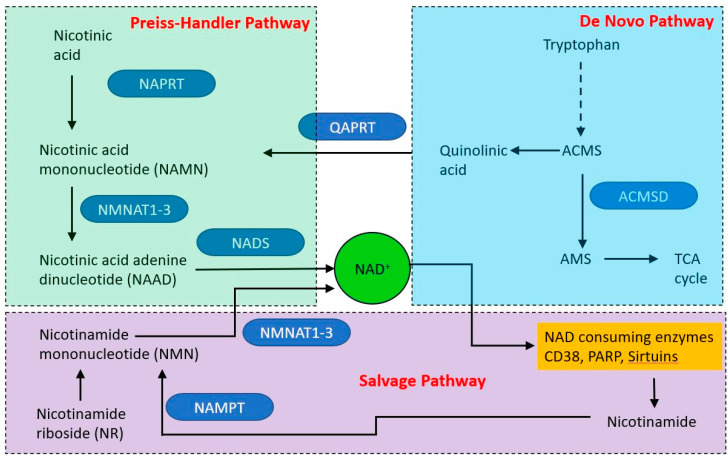
Overview of nicotinamide adenine dinucleotide (NAD) synthetic pathways. Three major pathways produce NAD, two from dietary starting points and one (the salvage pathway) from nicotinamide derived from NAD catabolic pathways. Single step reactions are indicated by solid arrows, the dashed arrow indicates a series of reactions.

**Figure 6 ijms-21-07580-f006:**
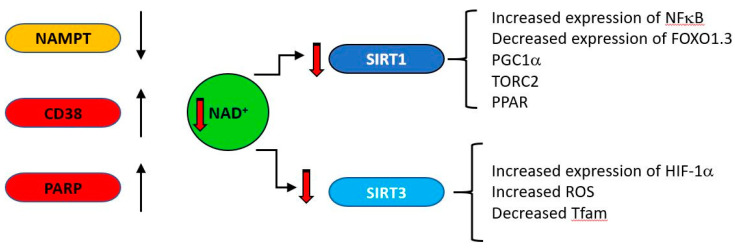
During ageing, upregulation of NAD consuming enzymes such as CD38 and poly (ADP-ribose) polymerase (PARP) and down regulation of the synthetic enzyme nicotinamide phosphor-ribosyltransferase (NAMPT) result in reduced levels of NAD. This is results in lower activity of the NAD–dependent deacylases (sirtuins). SIRT1 and SIRT3 regulate a variety of mitochondrially related functions whose age related change is regarded as deleterious, including decreases in transcription of genes such as PGC1a, PPAR, TFAM and FOXO1.3 and upregulation of NFκB.

**Figure 7 ijms-21-07580-f007:**
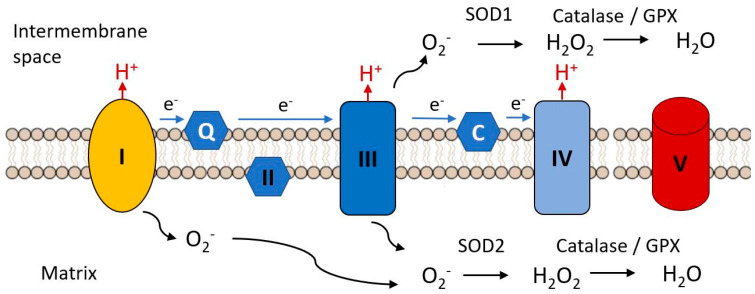
Complex I and III are the major sites of the electron transfer chain at which reactive oxygen species are released. The superoxide radical (.O_2_^−^) is the initial species generated by reduction of molecular oxygen. It is reduced to hydrogen peroxide by superoxide dismutases 1 or 2 and is further reduced to water by catalase or glutathione peroxidase, these steps constituting the initial anti-oxidant defense mechanisms. Black arrows indicate chemical transformations, blue arrows indicate electron flow.

**Figure 8 ijms-21-07580-f008:**
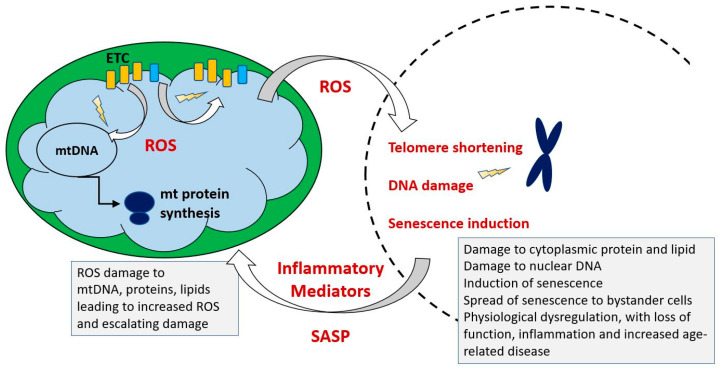
Summary of mechanism proposed by the mitochondrial free radical theory of ageing. Reactive oxygen species produced during operation of the electron transfer chain cause damage to mitochondrial components, including mtDNA, proteins and lipids. This results in accumulation of increasingly damaged components of the electron transport chain (ETC), resulting in exacerbated production of ROS, which are also able to damage extra-mitochondrial cellular components. Additionally, the increased ROS may trigger epigenetic changes, including telomere shortening and upregulation of cellular senescence pathways.

**Figure 9 ijms-21-07580-f009:**
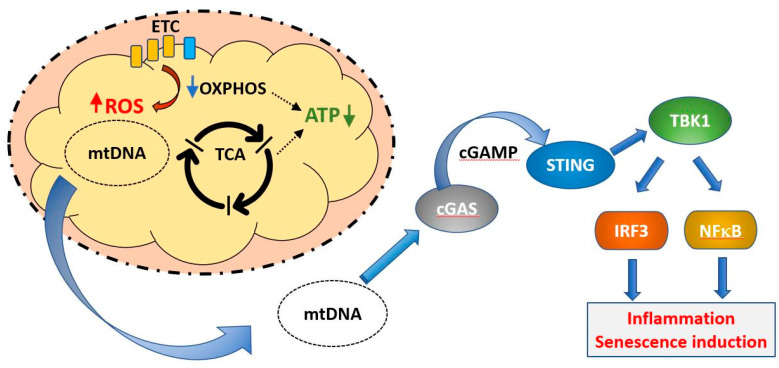
Induction of cellular senescence programs by mitochondrial DNA. Under normal healthy conditions, defective mitochondria (exhibiting decreased OXPHOS, blue arrow, decreased ATP production, green arrow, and increased ROS production, red arrow) are removed by mitophagy and mitochondrial DNA is not released into the cytoplasm. If this process is unable to deal with the increasing load of mitochondrial damage during ageing, mtDNA may be released to the cytoplasm, where it constitutes a damage associated molecular pattern (DAMP). mtDNA is recognized by the enzyme cGAMP, which synthesizes the intermediate cyclic guanosine monophosphate–adenosine monophosphate (cGAMP). This activates the transcription factor stimulator of interferon genes (STING), which upregulates transcriptional programs controlled by IRF3 and NFκB, resulting in the induction of cellular senescence. Blue arrows indicate the pathway connecting dysfunctional mitochondria with senescence induction.

**Figure 10 ijms-21-07580-f010:**
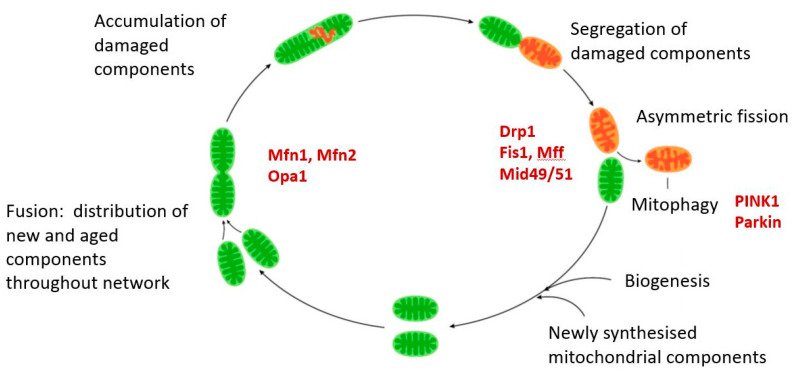
The mitochondrial life cycle. Newly synthesized components are added to existing mitochondria. These are shared throughout the mitochondrial pool by cycles of fission and fusion which constitute part of the mitochondrial life cycle. Fusion events are critically dependent on mitofusin s 1 and 2 and the inner membrane GTPase Optic Atrophy-1 (OPA1). Fission is initiated by recruitment of the cytosolic GTPase dynamin-related protein 1 (DRP1) to the mitochondrial surface, where it docks with one of several receptors, Fis1, mitochondrial fission factor (Mff), Mid 49 or Mid 51. Damaged molecules are segregated and asymmetric fission results on one daughter with normal mitochondrial membrane potential, the other containing the damaged molecular constituents and having a low membrane potential (MMP). This low potential triggers mitophagy by the recruitment of the PTEN-induced kinase 1 (PINK1), which then activates the E3 ubiquitin ligase Parkin. Ubiquitinated mitochondria are then disposed of by mitophagy. Additional pathways to trigger the removal of damaged mitochondria have been described and these may be initiated by signals other than reduced MMP, such as externalized cardiolipin.

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
