# Peer review of "Intimate Relations—Mitochondria and Ageing"

_ijms, 2020, doi:10.3390/ijms21207580_

Round 1

Reviewer 1 Report

While the topic is of outstanding interest, the review in its present form is not well structured, poorly informative, and difficult to read.

The manuscript does not contain figures and/or tables that could help the reader to visualize the most important notions. I suggest the authors to add some figures to summarize the major manifestations of mitochondrial age-related dysfunction and to graphically summarize the different paragraphs. I also suggest to include a paragraph regarding the relationship between mitochondrial proteases and aging in chapter 8 (Mitochondrial dynamics). In its present from, the review is unbalanced, and does not contain enough critical reasoning.

Author Response

We have reorganised the manuscript and hope this helps with clarity.  We have also added several sections on topics requested by the present and the other referees, and hope that this adds to the balance and informative value of the manuscript.

We thank the referee for the useful suggestion to add figures,  We have now included 10 figures to illustrate some of the key points that we make in the manuscript, and we hope this aids clarity.    Figure 1 is an overview of some of the main manifestations of ageing in mitochondria that we discuss in the text, and the remaining figures illustrate specific topics as we discuss them. 

We have added a section on mitochondrial proteases, but have included this in the biochemistry section (lines 510-575), as it seemed that they are involved in a wider range of activities than just dynamics.

Reviewer 2 Report

This i a systematic review of chnges in mitochondrial structure and function occurring in aging,, encompassing biochemical changes, genetics (mitochondrial and nuclear DNA), stem cell programming, quality control (mitophagy, fusion/fission(. The free radical theory of aging is discussed, and criticisms on its full validity are summarized. This excellent review is both comprehensive and critical, and is very useful for both specialists and non-specialists in the field.

There are however a few points t be clarified.

Biochemical changes of the OXPHOS system (complexes, supercomplexes, resèiration and >ATP synthesis), now spread in different sections, should be condensed in a subsection of biochemical changes. Concerning suprcomplexes, there are papers describing changes in aging, that should be quoted.

The criticisms to the Free Radical Theory of Aging have been disputed by Barja in a recent review. It would be fair to mention it.

Minor points.

Lines 106-107. Uniform ketoglutarate and oxoglutarate

Line 192. ..lower than the WT (?)

Line 546. ATP synthase is complex V

Lines 561-65. Not very clearly expressed

Author Response

We thank the referee for the positive evaluation .

In response to the suggestions made, we ha consolidated the references to OXPHOS and energetics in a single section, and added a section on supercomplexes (beginning line 212 of the revised manuscript).  We thank the referee for this suggestion and feel it enhances the discussion.

We have added a short paragraph on Barja’s defense of the free radical theory of ageing, and taken this opportunity to reiterate our own view that, although the theory as originally stated is too simplistic, elements of it are important in amore nuanced appreciation of the role of ROS in ageing.

We have referred to oxoglutarate / ketoglutare uniformly in the text as “oxoglutarate”

We have corrected to “lower than WT

We have corrected the designation of ATP synthase to “complex V”.

The lack of clarity in original lines 561-565 probably arose because of a couple of sentences on the role of ubiquinone as an electron transporter.  As this was not critical to the point we were making, we have deleted them.

Reviewer 3 Report

This is a clearly written, comprehensive review on the numerous connections that exist between mitochondrial behavior and aging. I have no major concerns. I commend the authors for a thorough job. In the mitochondrial dynamics section, I, however, would like the authors to comment on:

i) Role of the Drp1 adaptor, Fis1, in mitophagy-related mitochondrial fission.

ii) Role of cardiolipin and its externalization to the outer mitochondrial membrane (under stress conditions) as a cue for mitophagy, and the probable role of cardiolipin in monitoring mitochondrial health and aging.

Author Response

We thank the referee for the positive evaluation. 

In response to the suggestions, we have added a paragraph on the role of FIs1 in mitophagy-related fission (lines 1063-1067 in the revised version).

We thank the referee for the suggestion to include a discussion of the role of cardiolipin in monitoring mitochondrial health.  We have added a new section on cardiolipin in the revised version (lines 575-600)

Round 2

Reviewer 1 Report

The revised version has been consistently ameliorated and has improved the review.

Just one minor point: the red color to indicate healthy mitochondria in figures is misleading. I would change it.

Author Response

Thanks you for the comments and for the original suggestions to improve the review.  We have revised the figure colour scheme consistently.  Healthy mitochondria are now in green/blue, sick mitochondria are yellow / red.
